# SMC modulates ParB engagement in segregation complexes in *streptomyces*

Katarzyna Pawlikiewicz[1], Agnieszka Strzałka ®[1], Michał Majkowski[2], Julia Duława-Kobeluszczyk[1], Marcin J. Szafran[1] & Dagmara Jakimowicz ®[1] ✉

ParB is a bacterial chromosome segregation protein with recently demonstrated CTPase activity. CTP-bound ParB homodimers are loaded onto DNA at *parS* sites and spread along DNA, forming a large nucleoprotein complex. ParB complexes recruit condensin (SMC protein). Whether SMC modulates ParB complexes has remained unknown. Here, we employ *Streptomyces venezuelae* strains producing ParB-HaloTag in the presence or absence of SMC and use single-cell time-lapse fluorescence microscopy, single molecule tracking and fluorescence recovery after photobleaching analyses to explore ParB dynamics. Additionally, we perform chromatin immunoprecipitation to examine ParB interactions with DNA, with or without SMC. We reveal that SMC modulates ParB complex stability and ParB mobility. We find that the absence of SMC reduces ParB spreading. Additionally, we show that SMC reduces ParB CTPase activity in vitro. Taken together our data provide evidence of SMC positive feedback on the ParB nucleoprotein complex, offering insight into the nature of ParB complexes.

The molecular mechanism of bacterial chromosome segregation is not fully understood. In numerous bacterial species, chromosome segregation is driven by the ParABS system[1–3]. ParB forms the nucleoprotein segregation complex (segrosome) by binding *parS* sites that are clustered in the proximity of the origin of chromosome replication (*oriC*)[3–6]. ParB complexes recruit structural maintenance of chromosomes (SMC) proteins, which play a key role in higher-order DNA organisation. This organisation, by ParB and SMC, facilitates the segregation of the *oriC* and its proximal region of the chromosome. To achieve this, segrosomes interact with ParA, an ATPase that fuels their movement[1,3,7,8]. The ParB complex stimulates the ATPase activity of ATP-bound ParA dimers that are nonspecifically associated with DNA, helping to release ParA from the nucleoid[9,10]. This interaction generates a gradient of nucleoid-bound ParA, which provides the driving force for chromosome segregation[11,12].

ParB was recently demonstrated to be a CTPase, opening new avenues for understanding segrosome assembly[13,14]. According to current models, ParB binds *parS* sites as a CTP-bound homodimer (referred to as the "nucleation stage") and subsequently undergoes a conformational change to form a closed clamp structure[13,15]. In this clamp conformation, ParB dimers slide along the DNA. ParB spreading away from the *parS* sites drives formation of higher-order DNA-ParB complexes[16]. CTP hydrolysis then releases ParB from the DNA, which is accompanied by a conformational change. CTP hydrolysis deficient ParB variants exhibited excessive spreading, indicating that nucleotide hydrolysis increases the turnover of segrosomes[17]. While the role of CTP binding and hydrolysis was established, some of the details of ParB complex formation on DNA are still elusive. For example, the impact of SMC loading on the ParB complex and its interaction with ParA remained unexplored.

Recruitment of SMC by the ParB complex was demonstrated in numerous bacterial species[9,18–20]. SMC acting as a loop extrusion motor promotes long-distance contacts and alignment of chromosomal arms[19,21–23]. Distorted chromosome organisation in the absence of SMC may lead to chromosome segregation and/or morphological defects[24]. In *Bacillus subtilis*, SMC was reported to directly interact with the N-terminal domain of ParB, which is also the domain engaged in ParA and CTP binding[25]. Interestingly, SMC recruitment by ParB depends on

[1]Department of Molecular Microbiology, Faculty of Biotechnology, University of Wroclaw, Wroclaw, Poland. [2]Advanced Imaging and Cytometry Laboratory, Faculty of Biotechnology, University of Wroclaw, Wroclaw, Poland. ✉e-mail: dagmara.jakimowicz@uwr.edu.pl

CTP binding but does not require CTP hydrolysis[17]. The interactions between SMC and ParB are proposed to affect ParB association with ParA[17,26]. The contribution of ParA to SMC loading was shown, indicating the crosstalk between both ParB's binding partners, ParA and SMC[27].

In *Streptomyces*, multigenomic, filamentous and sporulating bacteria, the ParB and SMC proteins organise multiple chromosomal copies in a developmental stage-dependent manner[28,29]. ParB binds to numerous (16 in *S. venezuelae*) *parS* sites located near the *oriC* region at the centre of the linear *Streptomyces* chromosome[30,31]. In apically growing and branching vegetative cells, the main role of ParB complexes is to anchor *oriC* of the apical chromosome to the cell pole (hyphal tip), thereby facilitating efficient tip extension and branching. This anchorage of tip-associated ParB complex requires ParA to interact with polar proteins (polarisome, also known as a tip-organising complex, TIPOC)[32–35]. During sporulation, which turns elongated sporogenic cells into chains of unigenomic exospores, multiple chromosomal copies are segregated and compacted. The extensive elongation of the sporogenic cell, accompanied by intensive chromosome replication, is followed by growth arrest. Next, the sporogenic cell undergoes multiple synchronised cell divisions. At this stage, ParA and ParB ensure efficient segregation of chromosomes into spores[30,36,37]. ParA accumulates along sporogenic cells while ParB forms an array of uniformly distributed segrosomes. Recently, it was shown that in *Streptomyces*, as in other bacteria, ParB complexes recruit SMC[28]. The SMC recruitment is required for efficient chromosome compaction within the spores. SMC loading by ParB rearranges *Streptomyces* chromosomes during sporogenic development from an open conformation with limited interarm contacts to a closed conformation with both arms aligned in spores[28].

Given that ParB complexes in *Streptomyces* recruit SMC, we wondered if SMC loading would impact the positioning and dynamics of ParB complexes. We expected that the loading of SMC, which is enhanced in sporogenic cells, would facilitate the progress of chromosome segregation. Therefore, we labelled ParB with HaloTag (ParB-HT) in *S. venezuelae*, the model species, which allows the application of time-lapse microscopy for analysis of sporogenic development. We followed the dynamics of ParB-HT complexes in sporogenic hyphae of *S. venezuelae* in the presence and absence of SMC. We found that the absence of SMC shortens the lifetime of ParB-HT segregation complexes but also reduces the mobility of ParB-HT. ParB-DNA binding studies in wild-type and *smc* mutant strains show that SMC promotes ParB binding to *parS* and spreading, while measurements of ParB CTPase activity demonstrate that SMC reduces CTP hydrolysis by ParB. Together, our findings reveal previously unknown features of ParB complexes, establishing that SMC loading has a positive feedback on segrosome stability.

## Results

### The dynamics of ParB complexes in sporogenic cells

To study the dynamics of ParB complexes during sporogenic chromosome segregation of *S. venezuelae*, we labelled ParB by fusing it with HaloTag (HT). The levels of ParB-HT were similar to the levels of wild-type ParB, either for ParB-HT produced from a repressible promoter or from the native *parAB* promoter (Δ*parB* p$_{rtet}$*parB-HT*, strain KP009, or Δ*parAB* p$_{nat}$*parAB-HT*, strain KP006, respectively, for details see the Supplementary Information). The ParB-HT fusion protein was confirmed to be functional and had no adverse effects on the growth of the KP009 or KP006 strains (Supplementary Fig. 1A, B). Fluorescence microscopy detecting HaloTag labelling with TMR ligand revealed the presence of irregularly spaced ParB-HT complexes in vegetative hyphal cells, with a characteristic bright focus at the hyphal tip representing the apical ParB complex (Supplementary Fig. 1C), consistent with previous analyses of *S. coelicolor*[33]. To determine the dynamics of ParB complexes during sporogenic development, the ParB-HT strain (KP009) was further modified to produce cell division

protein FtsZ fused with YPet (strain KP011) to mark Z-ring formation during cytokinesis, as described earlier[30,38].

The ParB-HT complex dynamics in sporogenic cells of the KP011 strain was followed with time-lapse single-cell fluorescence microscopy. ParB-HT was visible as irregularly spaced complexes during sporogenic cell extension (Fig. 1, Supplementary Figs. 2, 3 and Supplementary Movie 1). Shortly after cell extension stopped (T1 = 27 +/− 10 min), these distinct ParB-HT complexes became regularly spaced along the hyphal cell (resembling those observed earlier in *S. coelicolor*[39]) (Fig. 1, Supplementary Figs. 2, 3 and Supplementary Movie 1). Noticeably, for the next hour, the ParB complexes gradually disassembled and completely disappeared approximately 107 +/− 28 min after cell growth arrest (T2) (Fig. 1, Supplementary Fig. 2 and Supplementary Movie 1). Thus, regularly spaced segrosomes lasted approximately 80 min. The regularly spaced ParB-HT complexes accompanied Z-rings, which also formed approximately 27 +/− 12 min (T3) and were visible until 107 +/− 16 min (T4) after cell growth arrest (Fig. 1, Supplementary Fig. 2 and Supplementary Movie 1). Thus, ParB-HT complexes disassemble at the same time as Z-rings disassemble during *S. venezuelae* sporogenic cell division.

### Elimination of SMC promotes disassembly of ParB-HT complexes in sporogenic cells

We investigated whether SMC loading could affect segrosome dynamics. To test this, we compared the sporogenic development of the strain producing ParB-HT in the Δ*smc* genetic background (Δ*smc*Δ*parAB* p$_{nat}$*parAB-HT*, strain KP007) with the wild-type control (Δ*parAB* p$_{nat}$*parAB-HT*, KP006 strain) using single-cell time-lapse microscopy. Of note, ParB-HT (produced from p$_{nat}$*parAB-HT*) complemented chromosome segregation defects, detected in the Δ*smc*Δ*parAB* strain (Supplementary Fig. 1D).

In sporogenic cells lacking SMC (KP007 strain), regularly spaced ParB-HT complexes were detected somewhat earlier (T1 = 20 +/− 9 min from hyphae growth cessation) than in the wild type control (KP006 strain) (T1 = 29 +/− 9 min) (Fig. 2, Supplementary Fig. 4 and Supplementary Movie 2 and 3). Strikingly, the ParB complexes disassembled significantly earlier in the absence of SMC (T2 = 79 +/− 25 min from hyphae growth cessation) than in the control strain (T2 = 114 +/− 42 min) (Fig. 2, Supplementary Fig. 4 and Supplementary Movie 2 and 3). Notably, in the absence of SMC, the lifetime of ParB complexes was not only shortened but also showed lower variation than in the wild type background, while the distances between the segrosomes were not affected (Supplementary Fig. 3).

To determine the time of sporogenic cell division, ParB-HT-producing strains were labelled with fluorescent D-amino acid (NBD-amino-D-alanine, NADA) in the time-lapse microscopy analyses. In the wild type control strain (KP006), septation (NADA signal) was detected approximately 128 +/− 28 min (T5) after hyphal growth cessation and 14 +/− 32 min after ParB complex disassembly. The absence of SMC slightly delayed the time of septation (T5 = 147 +/− 32 min after cell growth cessation) compared with the control strain (Fig. 2). This was consistent with delayed Z-ring disassembly observed in the Δ*smc* background (Supplementary Fig. 5, Supplementary Movie 4 and 5). The overall time of sporulation (from growth arrest to detection of rounded spores, T6) was marginally extended by the absence of SMC (Fig. 2C and Supplementary Fig. 4).

The shortened lifetime of ParB-HT complexes in the absence of SMC could result from the lowering of ParB-HT levels. However, Western blotting of ParB-HT levels in KP006 and KP007 strains during their sporogenic development showed that they were close to constant during sporogenic development and that they were not affected by the elimination of SMC (Supplementary Fig. 6A–C). Thus, changes in ParB-HT levels cannot account for the more rapid disassembly of ParB complexes in the absence of SMC than in wild type. Overall, these

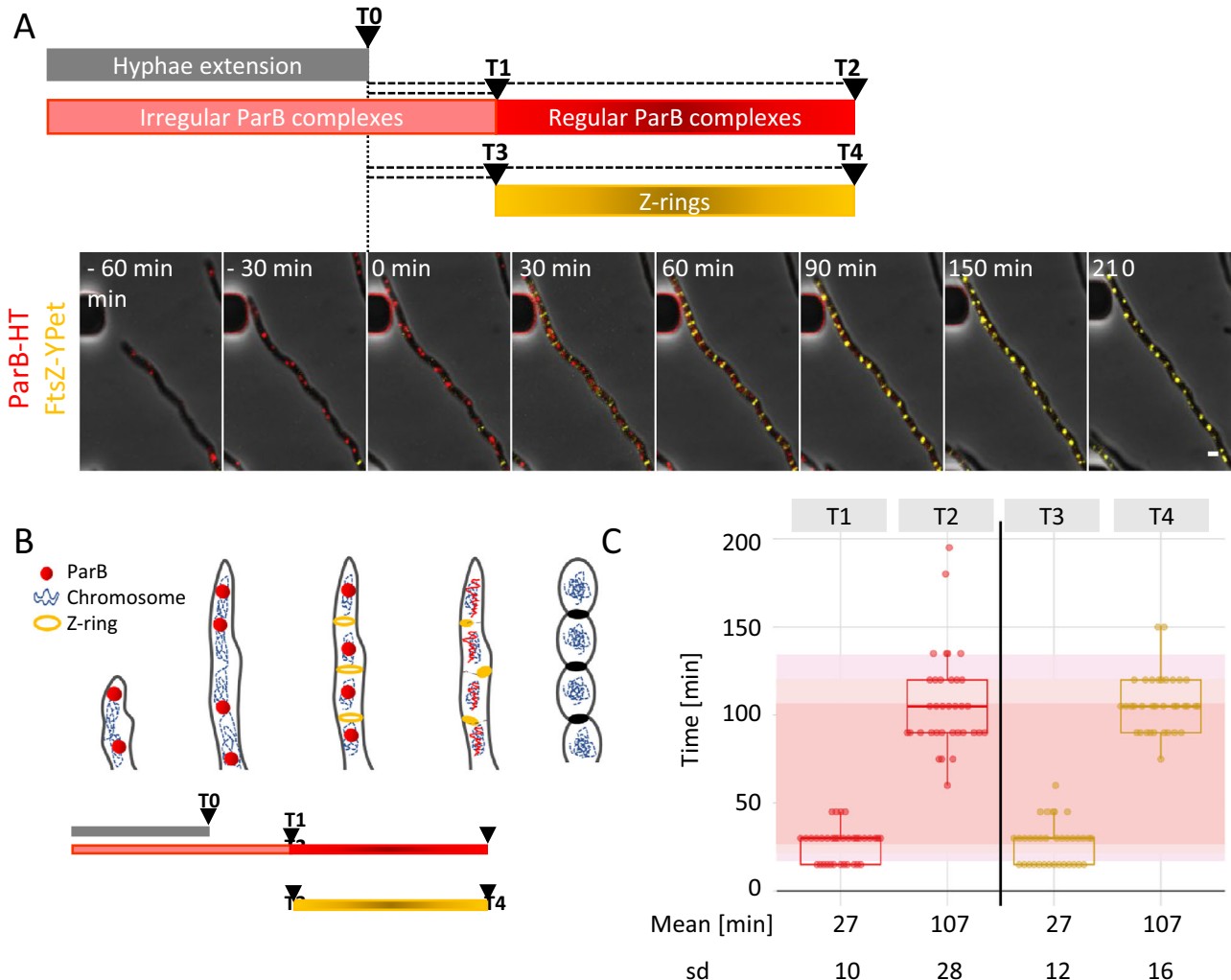

**Fig. 1 | ParB-HT complexes in _S. venezuelae_ sporogenic cells are short-lived and disassemble during cell division. A** Representative images from time-lapse analysis of sporogenic development of Δ*parB* p_rtet*parB-HT ftsZ-ypet* (KP011) strain showing fluorescence of ParB-HT stained with Janelia Fluor-549 (red) overlaid with FtsZ-YPet fluorescence (yellow) and phase contrast (grey) (Supplementary Movie 1). Time 0 is the time of sporogenic cell growth arrest, and the analysed stages of sporulation are indicated in the bar scheme below. Scale bar – 1 μm. For separate channels, see Supplementary Fig. 2. **B** Scheme of the sporogenic cell development indicating the stages which time was measured, the scheme also shows the changes of chromosome compaction, as described before[29]. **C** Analyses of the time elapsed from growth cessation (time 0) to appearance of regularly spaced ParB-HT complexes (T1), their disappearance (T2), appearance of regularly spaced Z-rings (T3) and their disappearance (T4). Red shading shows the mean lifetime of ParB complexes. Data shown in panel (**C**) were collected in 3 independent experiments for 35 hyphae. sd - standard deviation. Boxplots show the median value with 1st and 3rd quartiles as box boundaries and whiskers extending to mean +/− 1.5 * IQR. Source data are provided as a Source Data file.

findings indicate that SMC stabilises the ParB complexes during sporogenic development.

### Elimination of SMC lowers mobility of ParB molecules in sporogenic and vegetative cells

Next, we utilised single-molecule tracking to determine how *smc* deletion affects ParB-HT mobility. Earlier studies based on tracking various DNA-binding proteins demonstrated that increased DNA association often results in reduced mobility[40,41]. Accordingly, we expected that the disassembly of ParB-HT complexes would be reflected by increased mobility. ParB-HT mobility was studied at different stages of development: early sporogenic cells (17 h of liquid culture, where most of the sporogenic hyphae contained ParB-HT complexes), late sporogenic cells (22 h of liquid culture, where in some of the hyphae ParB-HT complexes disassembled) (Supplementary Fig. 7A), as well as in spores.

We found that the mobility of ParB-HT, determined as single-step distance (the distance that single molecules cover between two time points) raised during differentiation (Fig. 3A). Similarly, the average

apparent diffusion coefficient (D) increased from 0.052 μm²/s in the early stage of sporulation to 0.106 μm²/s in the late sporogenic cell (Fig. 3B). ParB-HT mobility was the highest in spores (D = 0.220 μm²/s) (Fig. 3A, B). Our analyses showed that ParB-HT molecules could be grouped into subpopulations based on their mobility - an immobile population (D = 0.02 μm²/s) that consists of molecules likely associated with DNA and a mobile population (D > 0.30 μm²/s) that contains the molecules likely freely diffusing. The fraction of mobile molecules increased significantly during sporogenic development and in spores constituted more than 80% of protein molecules (Fig. 3B). These changes in ParB-HT mobility are consistent with the observed segrosome disassembly.

In early sporogenic cells, *smc* deletion did not change the mobility of ParB-HT molecules (average diffusion coefficient 0.051 μm²/s in Δ*smc* background, KP007 vs 0.052 μm²/s in wild type control KP006 strain, respectively) (Fig. 3A, B). However, in late sporogenic cells (22 h of growth), the average diffusion coefficient was significantly lower (0.045 μm²/s) in the absence of SMC than in the wild

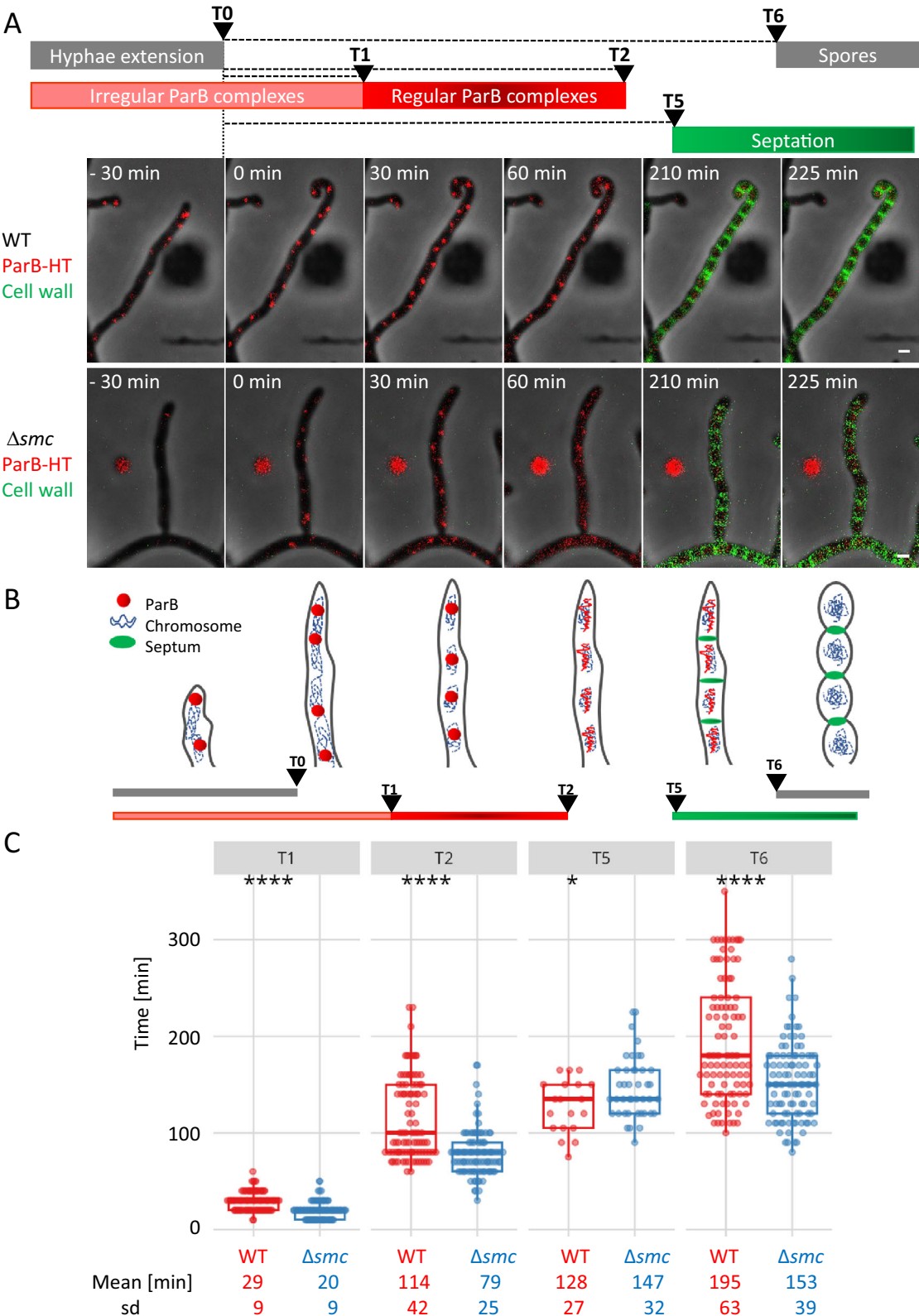

**C**

| T1 | T2 | T5 | T6 |
|---|---|---|---|
| **** | **** | * | **** |

| | WT | Δsmc | WT | Δsmc | WT | Δsmc | WT | Δsmc |
|---|---|---|---|---|---|---|---|---|
| Mean [min] | 29 | 20 | 114 | 79 | 128 | 147 | 195 | 153 |
| sd | 9 | 9 | 42 | 25 | 27 | 32 | 63 | 39 |

type control strain (0.106 µm²/s). This was reflected by the increased fraction of the immobile ParB-HT molecules in Δsmc cells (62% in Δsmc KP007 strain) as compared in wild type control strain (41.5% in KP006) (Fig. 3A, B). Notably, the lack of SMC did not change ParB-HT mobility in spores as compared to the wild type control (Fig. 3A, B).

The reduced ParB-HT mobility that could indicate increased DNA binding in late sporogenic cells lacking SMC was surprising because it

seemed to contradict the shortened lifetime of segrosomes that we observed in our single-cell time-lapse microscopy analyses. This could be explained by the delayed entry of the Δsmc (KP007) strain into sporulation and, therefore, longer prevalence of hyphae still undergoing sporogenic cell division and chromosome segregation at the 22 h time point. However, at this time point, the fraction of Δsmc sporogenic cells that showed disassembled ParB-HT complexes was

**Fig. 2 | The elimination of SMC accelerates disassembly of ParB-HT complexes in *S. venezuelae* sporogenic hyphae. A** Representative image from time-lapse analysis of sporogenic development of Δ*parAB* p$_{nat}$*parAB-HT*, KP006 strain and Δ*smc*Δ*parAB* p$_{nat}$*parAB-HT*, KP007 showing fluorescence of ParB-HT stained with Janelia Fluor-549 (red) overlaid with fluorescence of NADA green stained septa (green) and phase contrast (grey) (Supplementary Move 2 and 3). Time 0 is the time of hyphal cell growth arrest. Scale bar – 1 μm. **B** Scheme of analysed time intervals, the scheme also shows the changes of chromosome compaction, as described before[29]. **C** Comparison of the time elapsed from growth cessation (time 0) to appearance of regularly spaced ParB-HT complexes (T1), their disappearance (T2), the appearance of septa (NADA signal) (T5) and spores (T6) in the wild type control and Δ*smc* background (KP006 and KP007). Data shown in panel (**C**) were collected in 3, or 2 for NADA stained hyphae, independent experiments for 96 hyphae of KP006 strain and 99 hyphae o KP007 strain (to determine T1, T2, T6) and 19 hyphae of KP006 strain and 43 hyphae of KP007 strain (to determine T5), statistical analyses were performed using a two-sided Student's *t* test. *p*-values: T1: 7.479e-12, T2: 2.365e-11, T5: 0.023, T6: 8.957e-08. sd - standard deviation. Boxplots show the median value with 1st and 3rd quartiles as box boundaries and whiskers extending to mean +/− 1.5 * IQR. Source data are provided as a Source Data file.

only marginally lower than in the wild type control KP006 strain (Supplementary Fig. 7B). Moreover, since sporogenic cell division and chromosome segregation follow the shutdown of chromosome replication[28], we performed marker frequency analyses to determine *ori:arm* ratio in both strains. This analyses showed no significant difference in chromosome replication rate between the wild type control (KP006) and Δ*smc* (KP007) strains during sporogenic development (Supplementary Fig. 7C). This suggests that minor differences in development did not account for the significant difference in ParB-HT mobility observed in the absence of SMC.

To further explore the impact of SMC on ParB mobility, we tested if *smc* deletion could modulate ParB mobility in early vegetative cells. Western blotting analyses showed that FLAG-SMC levels are constant throughout the life cycle of *S. venezuelae* (Supplementary Fig. 6D). At the early stage of growth (5 h), hyphal cells contain only a few chromosomes (usually less than 10 ParB foci), and the apical ParB complex is responsible for *oriC* anchorage at the hyphal cell tip. This apical ParB complex remains at a constant distance from the cell pole (~ 1.3 μm)[33] allowing for a detailed examination of its dynamics (Supplementary Fig. 8A). In early vegetative cells, as in sporogenic cells, the absence of SMC significantly reduced the mobility of ParB-HT (D = 0.055 μm$^2$/s in Δ*smc* KP007 strain compared to D = 0.124 μm$^2$/s in the wild type control KP006 strain). Consequently, the fraction of immobile ParB-HT molecules increased (62.3% in the Δ*smc* KP007 strain compared to 40.2% in the wild type control KP006 strain) (Fig. 3C, D). Notably, the impact of SMC was less pronounced for tip distal ParB mobility than for tip-proximal molecules (Supplementary Fig. 8B). Next, we determined the position of immobile molecules (mean square displacement, MSD 0–0.13 μm$^2$) in relation to the cell pole. In the wild type control strain, the immobile ParB-HT molecules were constrained to a single location of the cell (~1.5 μm from the hyphal tip), while in the absence of SMC the immobile ParB-HT were more dispersed (Fig. 3E). These results indicate that SMC lowers ParB-HT dynamics at the poles of vegetative cells, but immobile molecules are not engaged in the formation of the single ParB-HT complex.

## The absence of SMC increases the turnover of ParB-HT in the apical complex

We next analysed the dynamics of the apical ParB-HT complex in wild type control and Δ*smc* strains (KP006 and KP007, as above). Since this complex is present throughout the vegetative growth and does not disassemble, to determine its stability and ParB-HT turnover, we used fluorescence recovery after photobleaching (FRAP). While in both analysed strains (KP006 and KP007) about 60% of ParB-HT complex fluorescence recovered after photobleaching, the time required for half-recovery (tau) was shorter in the absence of SMC (on average 135 +/− 72 s in KP007) than in wild type control strain (on average 222 +/− 200 s in KP006) (Fig. 4, Supplementary Fig. 9 and Supplementary Move 6 and 7). In addition, in the wild type control strain (KP006), the variation in recovery time was much higher than in the absence of SMC. Shortened recovery time indicates that a lack of SMC increases the availability of ParB-HT molecules for complex restoration, indicating faster turnover. The observed higher complex turnover rate in the absence of SMC corresponds to the lowered stability of

segrosomes in sporogenic cells but contradicts the decreased mobility of ParB-HT determined by single-molecule tracking.

## SMC promotes ParB spreading at *parS* sites

Having found that the absence of SMC reduces the mobility of ParB-HT but also lowers the stability of ParB-HT complexes, we set out to examine whether the observed differences could be attributed to SMC-mediated modulation of ParB interactions with DNA. To this end, we analysed the binding of ParB (untagged protein) to DNA in the wild type and Δ*smc* strains using ChIP-seq with anti-ParB antibodies (using the Δ*parB* strain, MD020, as a negative control). To eliminate any influence of the potential difference in the sporogenic development stage and because single-molecule tracking showed significantly increased ParB mobility in young vegetative cells lacking SMC (5 h of growth), our analyses were performed at this stage.

ChIP-seq analyses confirmed the specific binding of ParB to 16 *parS* sequences (as reported before[30]) and showed ParB binding to one additional *parS* site (Fig. 5A). In the wild type strain, ParB clearly bound to the regions adjacent to *parS* sequences extending bidirectionally up to 2.5-3 kb (Fig. 5A), which is consistent with the phenomenon of ParB spreading when in closed clamp conformation. Notably, in the Δ*smc* strain (TM010), ParB binding to *parS* was reduced, with the average number of reads two times lower than observed in the wild type strain (Fig. 5A–C). Moreover, in the absence of SMC, ParB binding adjacent to *parS* sites was almost completely eliminated and comparable to the control Δ*parB* strain (Fig. 5A). However, thorough analyses of ParB-DNA binding over the whole chromosome showed increased non-specific interactions of ParB in the absence of SMC as compared to the wild type strain (Fig. 5D). Thus, this indicates that the absence of SMC diminishes ParB association with *parS* and reduces ParB spreading from *parS* sites, however enhances non-specific ParB association with DNA. Lowered ParB spreading corresponds to lower complex stability, while its association with non-specific DNA explains the reduced mobility of ParB-HT in the absence of SMC.

## SMC reduces ParB CTPase activity

The spreading of the *B. subitilis* ParB homologue was shown to be regulated by CTP binding and hydrolysis[14,16,17,42]. Therefore, we sought to determine if the reduced *parS*-specific binding and spreading of ParB in the absence of SMC, as indicated by our ChIP-seq data, could be explained by the modulation of its CTP hydrolysis activity by SMC. We undertook a CTP hydrolysis assay measuring ParB CTPase activity in the presence of *parS*-containing plasmid DNA or the same plasmid DNA with mutated *parS* or in the absence of DNA, as well as in the presence or absence of purified FLAG-SMC (Supplementary Fig. 10A). This revealed that *S. venezuelae* ParB CTPase activity required the presence of *parS* as in the absence of DNA or in the presence of mutated *parS* it was very low, consistent with data obtained for other ParB homologues[13,14] (Fig. 6). Remarkably, the addition of FLAG-SMC to ParB (1:100 molar ratio) in the presence of *parS*-containing DNA significantly reduced CTP hydrolysis - from 65.1 +/− 2.9 μmol of phosphate released by 1 μmol of ParB in one hour in the absence of FLAG-SMC to 35.2 +/− 5.2 μmol of phosphate released in presence of FLAG-SMC (Fig. 6). Thus, SMC decreased ParB CTPase activity,

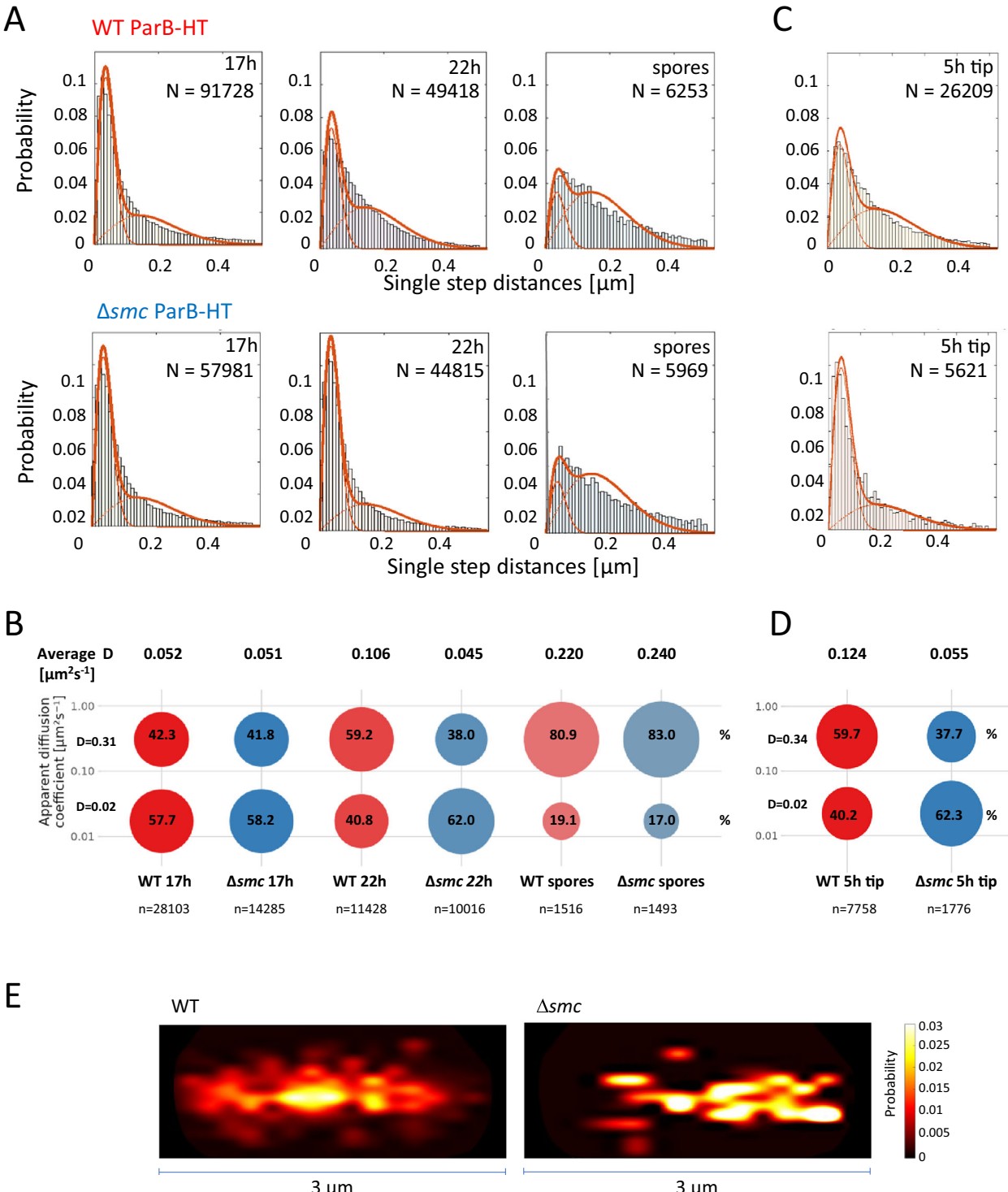

**Fig. 3 | Mobility of ParB-HT is lowered by the absence of SMC. A** Single-step distance of ParB-HT in sporogenic cells of the wild type control (KP006) and Δ*smc* strains (KP007) at different stages of sporogenic development. **B** Percentage of ParB-HT tracks with low (D = 0.02 μm²/s) and high (D = 0.31 μm²/s) diffusion coefficient at different stages of sporogenic development. **C** Single-step distance of ParB -HT analysis in young vegetative cells (3 μm distance from the tip) of the wild type control (KP006) and Δ*smc* strains (KP007). **D** Percentage of ParB-HT tracks with low (D = 0.02 μm²/s) and high (D = 0.34 μm²/s) diffusion coefficient in young vegetative cells (3 μm distance from the tip) of the wild type control (KP006) and Δ*smc* (KP007) strains. **E** Heatmap showing a number of tracks with apparent diffusion MSD between 0–0.13 μm² within 3 μm tip proximal region of young vegetative cells of the wild type control and Δ*smc* strains. Data shown in (**A**–**D**) were collected in 2 independent experiments for wild type control and Δ*smc* strains (Δ*parAB* p$_{nat}$*parAB-HT*, KP006 and Δ*smc*Δ*parAB* p$_{nat}$*parAB-HT*, KP007, respectively). **A**, **C** - Two population model fit and number of analysed steps are shown on each histogram. **B**, **D** - Average apparent diffusion coefficient (**D**) and number of analysed tracks (n) are shown for each strain. Source data are provided as a Source Data file.

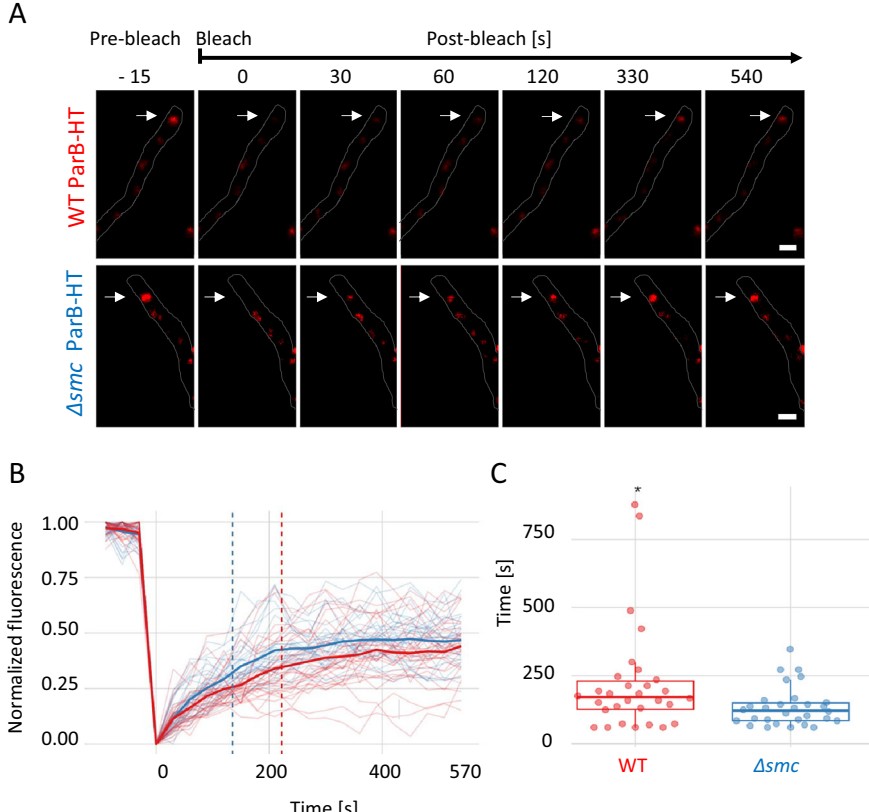

**Fig. 4 | Elimination of SMC shortens the ParB-HT complex recovery time after photobleaching. A** Representative image showing the photobleaching of the ParB-HT complex stained with Janelia Fluor-549 in young vegetative cells of the wild type control and Δ*smc* strains (Δ*parAB* p$_{nat}$*parAB-HT*, KP006 and Δ*smc*Δ*parAB* p$_{nat}$*parAB-HT* KP007, respectively). ParB-HT fluorescence (red) with hyphal cell contour overlaid (for separate chanells see Supplementary Fig. 9A, Supplementary Movie 6 and 7). **B** Fluorescence recovery analysis. Intensity of fluorescence plotted against the time of analyses. The dotted lines show mean Tau values for each strain.

**C** Tau – recovery half time - the time required for recovery of half fluorescence intensity calculated for the wild type control and Δ*smc* strains (Δ*parAB* p$_{nat}$*parAB-HT*, KP006 and Δ*smc*Δ*parAB* p$_{nat}$*parAB-HT* KP007, respectively). Data shown in B and C were collected in 3 independent experiments for 30 ParB-HT complexes of each strain. Statistical analysis was conducted using a two-sided Wilcoxon test, p-value: 0.036. Boxplots show the median value with 1st and 3rd quartiles as box boundaries and whiskers extending to mean +/− 1.5 * IQR. Source data are provided as a Source Data file.

which is consistent with the observed enhanced ParB spreading along DNA.

To determine whether the direct interaction between ParB and SMC is necessary for the observed influence of SMC on ParB's CTPase activity, we constructed a ParB variant that is impaired in SMC binding. This ParB variant, referred to as ParB$_{SMC-}$, was based on prior studies involving *B. subtilis* ParB[25]. Notably, two of the three amino acids identified as being directly involved in the interaction with SMC (L69 and K70 in *B. subtilis*) are located in a non-conserved loop between two β-structures. In *S. venezuelae*, this loop is slightly longer by one amino acid (Supplementary Fig. 10B, C). To create the *S. venezuelae* ParB$_{SMC-}$ variant, three amino acids in this loop TDR 132–134 were substituted with AAT, and an additional exchange D164R was also introduced (equivalent to E101K in *B. subtilis*) (Supplementary Fig. 10C). The ParB$_{SMC-}$ variant was then purified as a GST-fusion protein, similarly to the wild type ParB (Supplementary Fig. 10D, Supplementary Information).

In the CTPase activity assay, the ParB$_{SMC-}$ variant exhibited reduced CTP hydrolysis compared to wild type ParB (46.7 +/− 3.9 μmol of phosphate released by 1 μmol of ParB in one hour) (Fig. 6). This reduced activity is likely due to the modifications being in close proximity to the CTP-binding motif (GERR 140–143) (Supplementary Fig. 10B, C). Nevertheless, the activity of ParB$_{SMC-}$ remained dependent on DNA containing the *parS* sequence, confirming that it still interacts with DNA (Fig. 6). Notably, in the presence of FLAG-SMC (1:100 molar ratio), the CTPase activity of the ParB$_{SMC-}$ variant was similar to that observed in the absence of SMC (39.0 +/− 4.5 μmol of phosphate

released by 1 μmol of ParB in one hour) (Fig. 6). The effect of SMC on the ParB$_{SMC-}$ variant was significantly weaker than that on wild type ParB. Thus, the analysis of this ParB variant emphasises the importance of the SMC interface for CTP hydrolysis and confirms that direct interactions between ParB and SMC are necessary for SMC's influence on ParB's CTPase activity, spreading, and complex stability.

## Discussion

In summary, our data provide fundamental insight into understanding segregation complex assembly. Our experiments reveal the impact of SMC on ParB complexes, which has not been characterised before. We found that the absence of SMC shortened the lifetime of ParB complexes on DNA and increased their turnover rate in *S. venezuelae* cells, as determined by time-lapse microscopy and FRAP analyses, respectively. These data suggest that SMC loading stabilises ParB complexes on DNA. Moreover, in cells lacking SMC, diminished binding of ParB to *parS* sites and reduced spreading was demonstrated by ChIP-seq. The diminished ParB spreading in the absence of SMC explains the lowered complex stability. Surprisingly, single-molecule tracking showed that the absence of SMC significantly lowered the mobility of ParB molecules, which seemingly contradicts the lowered complex stability. However, the immobile ParB molecules were dispersed throughout the cell, most likely due to increased non-specific interaction with DNA, which was confirmed by ChIP-seq. These molecules may facilitate the fast reassembly of the ParB complex, which was observed in the absence of SMC in the FRAP experiment. Finally, the presence of SMC

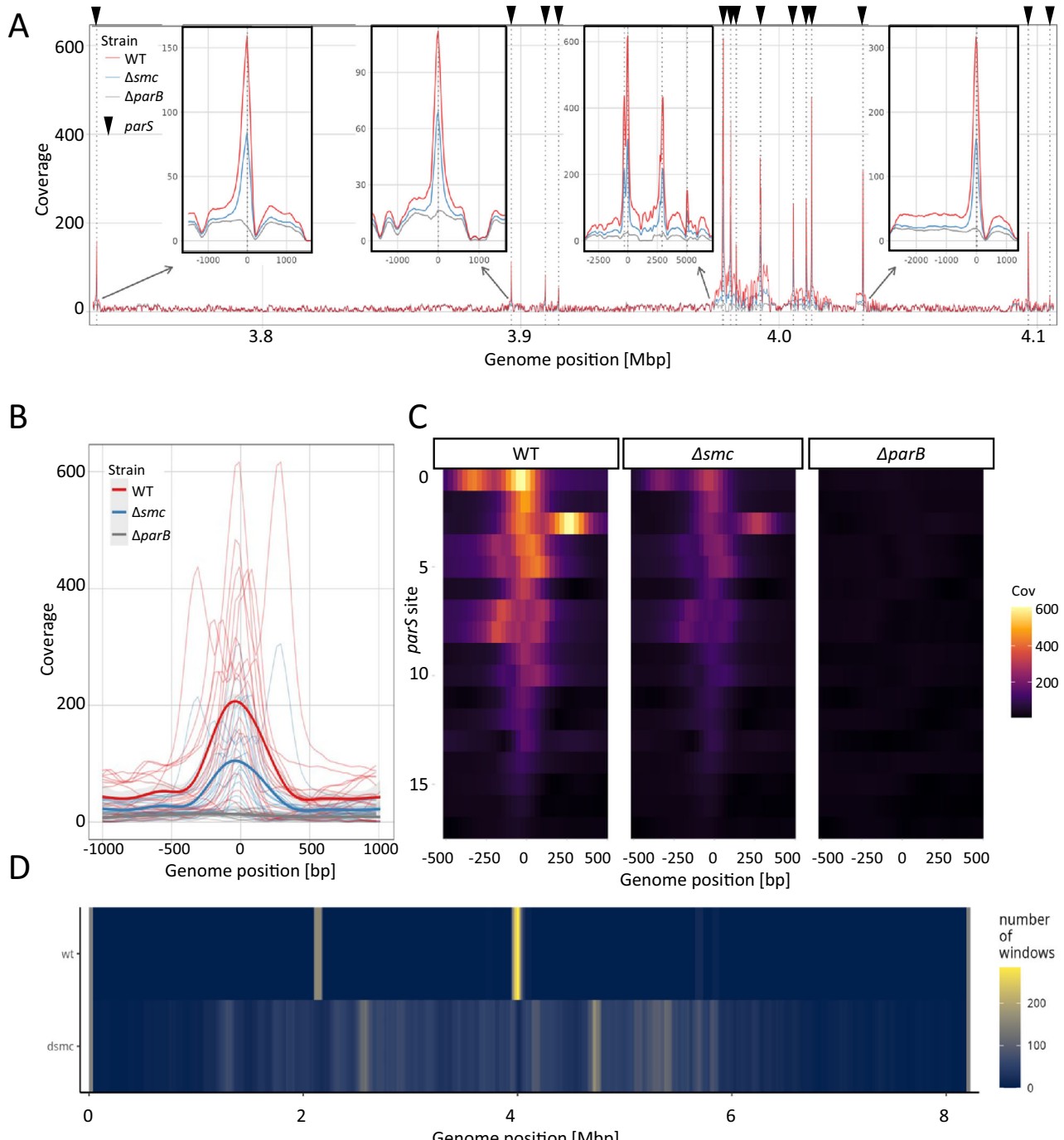

**Fig. 5 | ChIP-seq analyses showing diminished ParB-*parS* binding and spreading in the absence of SMC.** ChIP seq was performed using antigen-purified polyclonal ParB antibody and wild type, Δ*parB* (MD020) and Δ*smc* strain (TM010). **A** ChIP-seq detected binding plotted against the chromosomal region of *S. venezuelae* chromosome containing 17 *parS* sites in wild type (red), Δ*smc* (blue) and Δ*parB* (grey) strain. Insets show selected *parS* sites. **B** The average ChIP-seq signal at *parS* sites and neighbouring regions in strain wild type (red), Δ*smc* (blue) and Δ*parB* (grey). **C** Heatmap of 17 *parS* sites bound by ParB ordered by the number of reads detected at *parS* site in the wild type strain. **D** Average number of 100 bp long regions with enriched ParB binding in strains: wild type and Δ*smc*. Regions were counted using a 10000 bp rolling window. Source data are provided as a Source Data file.

reduced ParB CTPase activity in vitro, further corroborating with SMC promoting ParB spreading and stabilising the complex. The analysis of the ParB variant confirms that the direct interaction between ParB and SMC is required for the diminishing of ParB CTPase activity. Therefore, our data reveal positive feedback of SMC on the ParB complex stability (Fig. 7).

It was shown that upon the nucleation stage, i.e., binding of open ParB dimer to *parS* and CTP, the protein undergoes a conformational

change to form a closed clamp that facilitates ParB spreading[15,16,43]. While SMC recruitment to the ParB complex is well documented[9,18,20], the ParB conformation that recruits SMC has not been determined. It remains to be elucidated if SMC is recruited to ParB at the nucleation stage or to the ParB in a closed clamp conformation. Given the enhanced ParB binding to *parS* and spreading in the presence of SMC and taking into account that *Streptomyces* ParB (as other homologues) spreading requires CTP loading[16,17,42,44], we exclude the possibility that

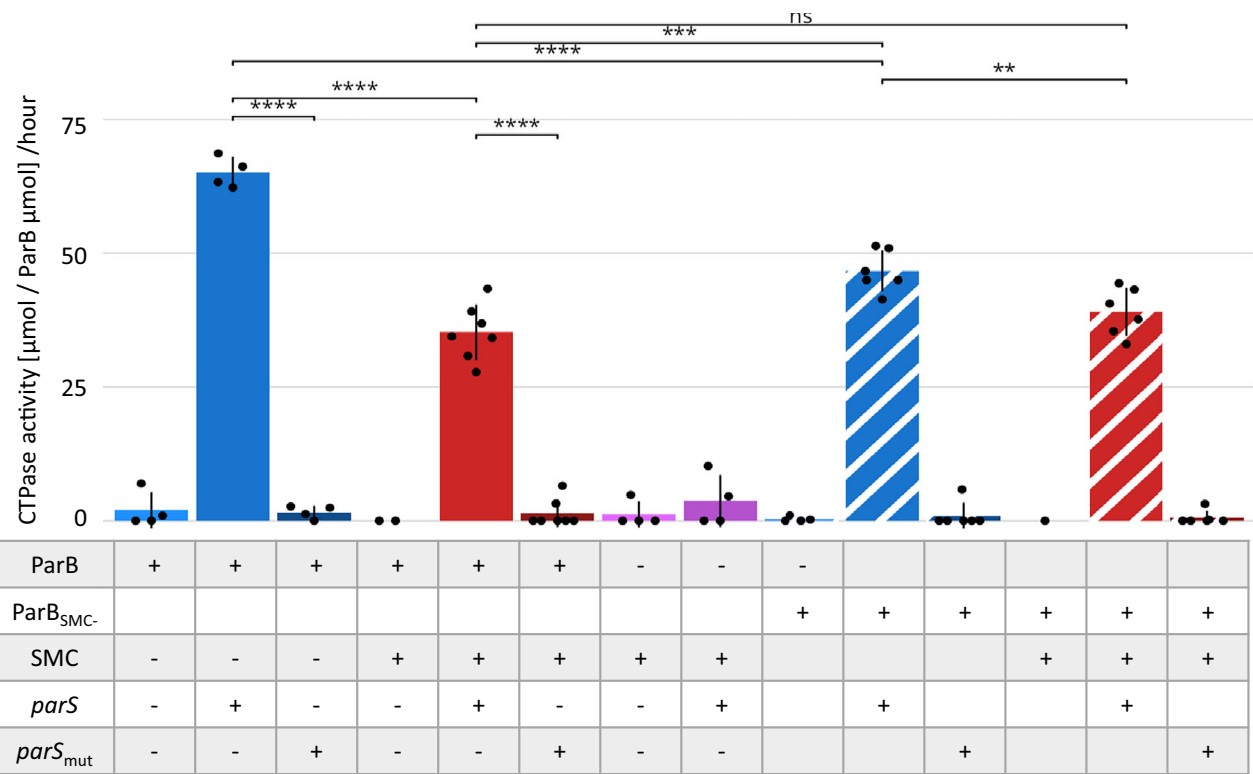

**Fig. 6 | SMC complex reduces ParB CTPase activity.** Bar plot mean +/− SD showing the results of CTP hydrolysis assay (phosphate detection) for ParB (2 μM), ParB (2 μM) in the presence of pBSK plasmid with single *parS* (15 μM), or ParB (2 μM) in the presence of pBSK plasmid with mutated *parS* (*parS*mut) (15 μM) in the absence or the presence of FLAG-SMC (0.02 μM), as compared to ParBSMC- (2 μM) variant in the presence of pBSK plasmid with single *parS*, or in the presence of pBSK plasmid with mutated *parS* (*parS*mut), in the absence or in the presence of FLAG-SMC (0.02 μM) (as indicated). Data were obtained from at least 4 independent experimental repeats. A two-sided Student's *t* test with Holm correction for multiple testing was used in statistical analysis. p-values: ParB_parS − ParB: 2e-16, ParB_parS − ParB_parS_SMC: 2e-16, ParB_parS_SMC − ParB_SMC: 2.2e-16, ParBSMC-_parS − ParBSMC-_parS_SMC: 0.0097, ParB_parS_SMC - ParBSMC-_parS_SMC: 1, ParB_parS - ParBSMC-_parS: 9.0e-10, ParB_parS_SMC - ParBSMC-_parS: 2.6e-02. Source data are provided in a Source Data file.

SMC inhibits CTP biding by ParB. Taking into account our finding that SMC promotes ParB binding and spreading, we should consider two alternative explanations. Either interaction with SMC facilitates ParB loading on *parS* and promotes a change to clamp conformation, or SMC stabilises ParB in closed clamp conformation. In the first scenario, the absence of SMC would be expected to reduce ParB spreading, which is indeed what we observe. However, if SMC promotes the ParB loading, one might expect an increase in the number of ParB molecules bound to DNA, leading to a greater number of molecules capable of hydrolysing CTP. This should result in an increased release of inorganic phosphate in the presence of SMC, which contradicts our observation of reduced inorganic phosphate release by ParB when SMC is present. This brings us to the alternative explanation: that SMC recruitment stabilises the closed clamp conformation of ParB. The inhibitory effect of SMC on CTP hydrolysis by ParB may result from either an inhibition of hydrolytic activity per se or from an inhibition of clamp opening, which would prevent ParB from dissociating from DNA and releasing hydrolysed CDP and inorganic phosphate. Moreover, the exchange of amino acid residues in ParB that are likely involved in direct interaction with SMC has been shown to diminish CTP hydrolysis and eliminate the impact of SMC on this process. Based on the available data, we cannot definitively determine whether SMC stabilises the clamp conformation of ParB or directly affects hydrolysis. To synthesise the experimental evidence we have provided and to explain the impact of SMC on ParB at substoichiometric concentrations, we propose that SMC recruitment by ParB both promotes further loading of ParB onto *parS* and stabilises the ParB molecules spreading along the DNA. Our model suggests that upon recruitment by ParB, SMC may enhance the binding of ParB molecules to distant regions of DNA by extruding DNA loops. These loops could also promote long-distance interactions between ParB molecules as they spread along the DNA. Such in-trans recruitment of ParB and the long-distance interactions that stabilise the complex have been previously described[42,45–47]. Thus, the binding of ParB to SMC-extruded DNA loops is expected to contribute to the stabilisation of the ParB complex, potentially by promoting interactions via the N-terminal domains (Fig. 7).

The surprisingly lowered mobility of ParB molecules in the absence of SMC, determined by single-molecule tracking, seemingly contradicts the lowered complex stability. Importantly, we observed that ParB mobility was highly elevated in spores when ParB complexes were disassembled, reflecting the ParB release from DNA. This is also consistent with earlier studies showing increased ParB mobility in the absence of *parS*[15]. One possible explanation of reduced ParB mobility in the absence of SMC, despite diminished *parS* binding, may be ParB ability to form condensates in a non-DNA bound state. The increased dissociation of ParB from *parS* could promote condensate assembly. While some studies suggest that ParB condensates require *parS* sequences, the formation of ParB clusters in the absence of *parS* was also postulated[48–51]. In addition, the ParB molecules released from *parS* in the absence of SMC may become engaged with other structures, such as nucleoid-bound ParA. Our global analyses of ParB binding showed protein association with non-specific DNA, supporting this explanation. Moreover, ParA was reported to influence ParB-SMC interaction, which indicates the crosstalk between both ParB partners[26,27]. It should be considered that ParA was shown to be present at the tips of young vegetative cells and along the sporogenic cells[30,33], where we observed lowered ParB mobility in the absence of SMC. Although this hypothesis requires additional investigation, our

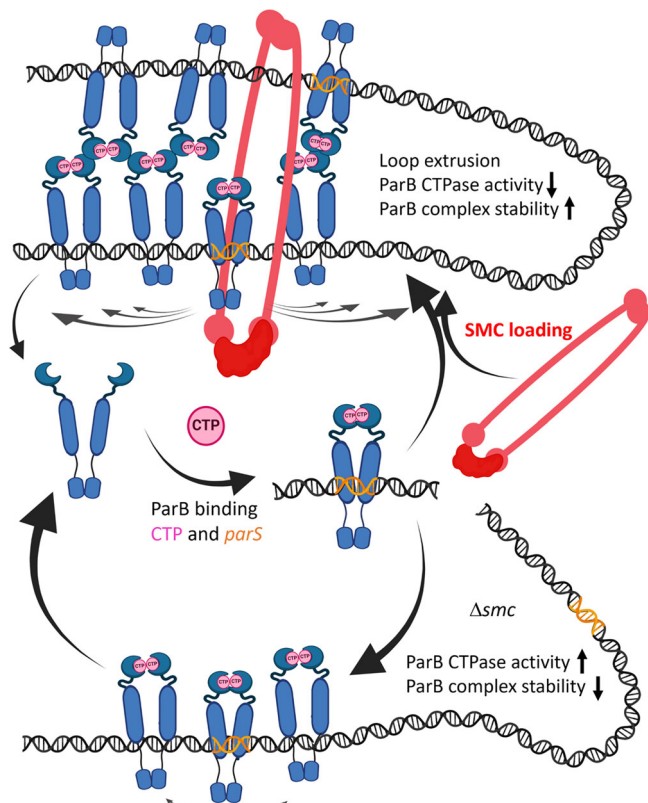

**Fig. 7 | Positive feedback of SMC loading on ParB complex assembly.** ParB binds *parS* as the CTP-bound dimer, which is followed by a change in the conformation to the closed clamp that slides away from *parS*. SMC loaded on the ParB complex extrudes DNA loops promoting ParB interactions that stabilise ParB in a clamp conformation, preventing clamp opening, inorganic phosphate release and ParB dissociation from DNA. ParB interactions on looped DNA stabilise the nucleoprotein complex. In the absence of the SMC ParB clamp, the DNA is destabilised and spreading is limited. Created in BioRender. Wolanski, M. (2025) https://BioRender.com/y0mo9g9.

observation nevertheless provides important insight into ParB regulation by SMC and the crosstalk between SMC and ParA.

In *Streptomyces*, ParB complexes play different roles depending on the growth stage – in sporogenic cells, they assist chromosome segregation, while in vegetative cells they anchor chromosomes at the tips of hyphal cells, facilitating efficient tip growth and branching. Using *Streptomyces* as the model provides some challenges due to their complex development and the presence of multiple chromosomes in the cell, but on the other hand, it offers a unique opportunity to study the disassembly of PaB complexes in sporogenic cells. Our results indicate that SMC stabilises ParB complexes in sporogenic cells undergoing synchronised cell division. At this stage, ParB complexes have been shown to recruit SMC, which leads to the rearrangement of the chromosome and juxtaposition of the chromosomal arms[28]. Intriguingly, we also observed that SMC stabilizes ParB complexes at the early vegetative stage. Little is known about the role of SMC at the earlier stages of growth. The earlier studies showed that in vegetative multigenomic *Streptomyces* cells, the chromosomes have predominantly open conformation with limited interarm contacts[28]. However, SMC expression is constant through the life cycle of *Streptomyces*, which suggests its involvement in chromosome organisation at all stages of growth. Markedly, ParB complex stability (determined by time-lapse microscopy and FRAP) was much more varied in the presence of SMC compared to its absence. We speculate that the variation of stability may be dependent on the progress or efficiency of SMC loading. It should be noted that time-lapse microscopy and FRAP

analyse ParB complex dynamics at different time scales. Nevertheless, the observed higher turnover rate of ParB-HT in the absence of SMC, as seen in FRAP, aligns with the less efficient spreading of ParB detected by CHIP-seq, both corresponding to lowered complex stability in a longer time scale. While our study demonstrates the role of SMC in ParB dynamics, the involvement of ParA in ParB complex assembly has been previously established[30]. Regardless, we cannot exclude the possibility that other developmental factors may also contribute to ParB dynamics.

Interestingly, in *S. venezuelae, smc* or *parB* deletion leads to modest chromosome segregation defects. However, both deletions when combined result in a significant increase of anucleate spores, which corroborates the results of earlier studies[52,53]. In *B. subtilis,* hydrolysis-deficient ParB in the absence of SMC resulted in severe chromosome segregation defects[17]. The disturbed chromosome segregation in the absence of SMC may be the consequence of disturbed chromosome condensation. In *Streptomyces*, SMC loading and chromosomal arm juxtaposition were shown to contribute to chromosome compaction, which allows regular distribution of the chromosomes along the sporogenic cell[28]. Given the hypothesis that the elimination of SMC is likely to affect the ParB-ParA interaction, the elimination of SMC could potentially disturb chromosome segregation by interfering with the ParB-ParA interaction. Thus, SMC may influence chromosome segregation by chromosome compaction, affecting ParB complex stability and its interaction with ParA, however, elucidation of the crosstalk between these three proteins calls for further studies.

In summary, our data provide fundamental insight into the understanding of segregation complex assembly (Fig. 7). Up to now, ParB was also shown to recruit SMC, however, the impact of SMC on the ParB complex has not been reported before. Our data prove the positive feedback of SMC on the ParB complex. The presented data set the stage for further mechanistic studies of ParB complex assembly in the presence of SMC. The structural studies are required to fully understand the conformational changes of ParB upon interaction with SMC and to elucidate the crosstalk between the three partners, ParA-ParB and SMC.

## Methods
### Bacterial strains and plasmids
Bacterial strains used in the study are listed in Supplementary Table 1, and the constructs are listed in Supplementary Table 2. Genetic manipulations were performed according to standard protocols and followed the specific enzyme or kit manufacturer's recommendations. *E. coli* culture conditions and manipulation methods followed the commonly employed procedures[54]. *S. venezuelae* strains were cultured in maltose yeast extract medium (MYM) agar plates or MYM liquid medium as described before[55]. Conjugation of plasmids and cosmids from *E. coli* ET12567/pUZ8002 to *S. venezuelae* was performed according to standard procedure, the conjugation mixture was plated on R2YE medium and flooded with appropriate antibiotics after 6 h of incubation[55]. The growth rate of *S. venezuelae* strains was analysed using a Bioscreen C instrument (Growth Curves US). Cultures (at least three independent cultures for the same strain, 300 µl culture volume) were set up by diluting spore suspension with $OD_{600} = 0.05$ 300 times in liquid MYM medium. The cultures grew for 48 h at 30 °C under the 'medium' speed and shaking amplitude settings, and their growth was monitored by optical density measurement ($OD_{600}$) every 20 min. The data were collected using BioScreener 3.0.0 software.

### Strains construction
The *S. venezuelae* strains used in this study are derivatives of the NRRL B-65442 strain described as the wild type. Primers used for the constructions are listed in Supplementary Table 3.

To construct a KP004 (Δ*smc*Δ*parAB*) and KP020 (Δ*parB*Δ*smc*) strain, the TM010 (Δ*smc*)[28] strain was modified. The exchange of *parB*

or *parAB* genes for a cassette containing the apramycin resistance gene and the *oriT* sequence was performed by intergeneric conjugation of the TM010 strain with *E. coli* ET12567/pUZ8002 containing the modified cosmid Sv-4-A09 Δ*parB* or Sv-4-A09 Δ*parAB*. The exconjugants sensitive to kanamycin and resistant to apramycin were selected. Obtained clones were verified by PCR with pSv_parABkont3_Fw/pSv_parABkont4_Rv primers, and PCR products were sequenced. In addition, strains were verified using Western blotting with anti-ParB antibody.

To construct the pKP03 (pSS170 p$_{rtet}$*halotag*), a 277 bp fragment encompassing p$_{tcp}$ promoter was amplified using KP_39Fw/KP_39Rv primers and pCRISPR-p$_{tcp}$-RBS$_{topA}$ plasmid as the template. The resulting fragment was cloned by the SLIC reaction to the pSS170*halotag* vector, digested with the XmaJI restriction enzyme. The reaction mixture was used to transform DH5α cells, and hygromycin-resistant transformants were selected. Next, the *tetRrev* gene (722 bp) was amplified using KP_37Fw/KP_37Rv primers and pTC-28S15-0X p$_{smyc}$*tetR*$_{rv}$p$_{tcp}$*topA*$_{Ms}$ as a template. At the same time, the fragment encoding the p$_{S14}$ promoter (120 bp) was amplified using KP_38Fw/KP_38Rv primers and pCRISPR-p$_{tcp}$-RBS$_{topA}$ as a template. An overlap PCR using KP_37Fw/KP_38.2Rv primers amplified both fragments: the *tetRrev* gene and p$_{S14}$ promoter, delivering an 801 bp product. The PCR product was cloned by SLIC to the pSS170-p$_{tcp}$*halotag* vector digested with the restriction enzyme HindIII. The reaction mixture was used to transform DH5α cells, and hygromycin-resistant transformants were selected. The obtained pKP03 (pSS170 p$_{rtet}$*halotag*) vector was verified by PCR (using pSSseq_Fw/pSSseq_Rv primers) followed by sequencing of the PCR product. The pKP03 (pSS170 p$_{rtet}$*halotag*) plasmid was introduced into wild type *S. venezuelae* (NRRL B-65442) by conjugation with *E. coli* ET12567/pUZ8002, followed by selection of hygromycin-resistant exconjugants. The obtained KP0003 strain was verified by PCR reaction (pSSseq_Fw/pSSseq_Rv primers) and Western blotting using an anti-HaloTag antibody.

The strains expressing *parAB-HT* under the control of the native promoter: KP005 (WT p$_{nat}$*parAB-HT*), KP006 (Δ*parAB* p$_{nat}$*parAB-HT*) and KP007 (Δ*smc*Δ*parAB* p$_{nat}$*parAB-HT*) were constructed using pKP05 (pSS170 p$_{nat}$*parAB-HT*) integrative plasmid. To construct the pKP05 plasmid, the sequence encompassing the *parAB* operon with its promoter (2587 bp) was amplified using KP_66Fw/KP_43Rv primers and cosmid Sv-4-A09 as a template. The product was cloned by SLIC to the pSS170-*halotag* vector, digested with XmaJI. The reaction mixture was used to transform *E. coli* DH5α cells, hygromycin-resistant transformants were selected. The obtained vector pKP05 (pSS170 p$_{nat}$*parAB-HT*) was verified by PCR (using pSSseq_Fw/pSSseq_Rv primers), followed by sequencing of the PCR product. The pKP05 vector was introduced to the wild type *S. venezuelae* (NRRL B-65442), MD030 (Δ*parAB*) and KP004 (Δ*smc*Δ*parAB*) strains by intergeneric conjugation with *E. coli* ET12567/pUZ8002, followed by a selection of hygromycin-resistant exconjugates. The obtained strains: KP005 (WT p$_{nat}$*parAB*-HT), KP006 (Δ*parAB* p$_{nat}$*parAB*-HT) and KP007 (Δ*smc*Δ*parAB* p$_{nat}$*parAB*-HT) were verified by PCR (pSSseq_Fw/pSSseq_Rv primers) and by Western blotting using anti-HaloTag and anti-ParB antibody.

In strains KP008 (WT p$_{rtet}$*parB-HT*), KP009 (Δ*parB* p$_{rtet}$*parB-HT*), KP010 (Δ*parB::apra* p$_{rtet}$*parB-HT*) and KP011 (Δ*parB* p$_{rtet}$*parB-HT, ftsZ-ypet*) *parB-HT* gene was under the control of p$_{rtet}$ promoter (p$_{tcp}$830 controlled by the reverse TetR repressor (TetR$_{rv}$) which binds to *tetO* sites in presence of tertracycline[56]). The strains expressing *parAB-HT* under the control of the p$_{rtet}$ promoter[56] were prepared using pKP08 (pSS170 p$_{rtet}$*parB-HT*). To construct pKP08, the sequence encoding the *tetRrv* gene (1068 bp) was amplified using KP_37Fw/KP_37BHTRv primers and pKP03 (pSS170 p$_{rtet}$*halotag*) plasmid as the template. At the same time, *parB* gene (1165 bp) was amplified using KP_43BHTFw/KP_43Rv primers and the Sv-4-A09 cosmid as a template. Both products were cloned to a pSS170 vector digested with the restriction

enzyme XmaJI using the Gibson assembly method. A reaction mixture was used to transform *E. coli* DH5α, and hygromycin-resistant transformants were selected. The obtained pKP08 (pSS170 p$_{rtet}$*parB-HT*) vector was verified by PCR reaction using pSSseq_Fw/pSSseq_Rv primers followed by sequencing of the PCR product. The pKP08 vector was then introduced to the *S. venezuelae* wild type (NRRL B-65442), MD020 (Δ*parB*), MD002 (Δ*parB::apra*) strains by intergeneric conjugation with *E. coli* ET12567/pUZ8002, followed by selecting hygromycin-resistant exconjugants. The obtained KP008 (WT p$_{rtet}$*parB-HT*), KP009 (Δ*parB*, p$_{rtet}$*parB-HT*) and KP010 (Δ*parB::apra* p$_{rtet}$*parB-HT*) strains were verified by PCR using pSSseq_Fw/pSSseq_Rv primers. To obtain the KP011 strain (Δ*parB*, p$_{rtet}$*parB-HT ftsZ-ypet*), the pKF351 was introduced to KP009 by intergeneric conjugation followed by a selection of apramycin-resistant exconjugants.

## Construction of pBSK plasmid containing *parS* and mutated *parS* site

To construct a pBSK plasmid containing a single *parS* site, the fragment of the *parAB* promoter region (365 bp) from *S. coelicolor* was amplified with primers parA_fw and parAB_rv, using chromosomal *S. coelicolor* DNA as the template. The obtained PCR product was digested with HindIII and BamHI and cloned into the pBSK vector digested with the same enzymes. The obtained pBSK*parS* vector was verified by digestion and sequencing. To introduce mutation in the *parS* site, PCR was performed using the following two pairs of primers: parA_fw with parSA_mut rv and parSA_mut fw with parAB_rv and pBSK*parS* as a template. Next, the obtained products were mixed to serve as the template for overlap PCR with primers parA_fw and parAB_rv. The obtained PCR product was digested with HindIII and BamHI and cloned into the pBSK vector, digested with the same enzymes. The obtained pBSK*parS*$_{mut}$ vector was verified by digestion and sequencing.

## Construction of the pGEX-*parB* and pGEX-*parB*$_{SMC-}$ vector and overproduction of recombinant *S. venezuelae* ParB and ParB$_{SMC-}$ protein

To construct the pGEX-*parB* vector, the sequence of the *parB* gene (*vnz_18015*) (1156 bp including 138 bp upstream the *parB* gene annotated in the StrepDB database, based on start codon determined by comparison with *S. coelicolor* ParB[31]) was amplified using a pair of oligonucleotides pGEX_parB_FW and pGEX_parB_RV and the chromosomal DNA of *S. venezuelae* as a template. The amplified DNA fragment was purified and cloned using the SLIC procedure to the pGEX-6P-2 plasmid digested with the restriction enzymes BamHI and XhoI. SLIC mixture was used to transform *E. coli* DH5α cells, and the ampicillin transformants were selected. The presence of the introduced sequence was confirmed by PCR, restriction digestion and the positive clones were verified by sequencing.

The pGEX-*parB*$_{SMC-}$ was constructed by the exchange of the *parB* gene in the pGEX-*parB* to *parB*$_{SMC-}$, that contained the exchange of the 7-nucleotide sequence (392–398 bp from start codon, introducing SnaBI restriction site), resulting in the exchange of TDR (132–134) to AAT and 3-nucleotide sequence in position 488–490, resulting in the exchange D164R. The fragment of *parB*$_{SMC-}$ gene (530 bp) encompassing the introduced mutation was custom synthesised (Thermo Fisher, USA) and cloned into pGEX-*parB* digested with PasI and HindIII restriction enzymes. The obtained clones were verified by digestion and sequencing.

For the overproduction of recombinant protein GST-ParB or GST-ParB$_{SMC-}$, 50 ml of LB liquid medium with the addition of ampicillin and chloramphenicol was inoculated with a single colony of *E. coli* BL21 (DE3) pLysS pGEX-*parB* or pGEX-*parB*$_{SMC-}$. Cultures were conducted overnight at 37 °C with shaking at 180 rpm. The next day, 800 ml of liquid LB medium containing ampicillin and chloramphenicol was inoculated by adding 16 ml of overnight cultures of *E. coli* BL21 (DE3)

pLysS pGEX-*parB* or pGEX-*parB_SMC*. Cultures was carried out at 37 °C with shaking at 180 rpm until an optical density of 0.5–0.6 was reached. Overproduction of GST-ParB protein was induced by the addition of 1 M isopropyl-β-d-1-thiogalactopyranoside (IPTG) solution to a final concentration of 0.5 mM. Cultures were continued for 3 hours, as described above. After this time, the cultures were centrifuged (10 min, 4000 g, 4 °C). The supernatant was discarded, and the cell pellet was suspended in 45 mL of 50 mM TrisHCl, pH8.0, 150 mM NaCl buffer. The cell suspension was sonicated and the resulting cell lysate was centrifuged (20 min, 25000 g, 4 °C). Next, the supernatant was filtered through a cellulose filter with a pore diameter of 0.45 μm. The filtered cell lysates were transferred to a 50 mL Falcon tub and 1 mL of Glutathione Sepharose® 4B chromatographic resin previously balanced with 50 mM TrisHCl, pH8.0, 150 mM NaCl buffer was added and incubated overnight at 4 °C, stirring constantly. The next day, the resin was transferred to a chromatography column. The packed column was washed with 100 mL of, 50 mL 50 mM TrisHCl, pH8.0, 150 mM NaCl buffer with 10% glycerol and next with 10 mL of Pres buffer (50 mM TrisHCl, pH 8.0, 150 mM NaCl, 1 mM DTT and 1 mM EDTA). To cleave GST protein from the GST-ParB protein 20 μl of PreScission protease (Thermo Fisher, USA) in 5 mL of Pres buffer was added to the resin. The resin was incubated at 4 °C overnight. The next day, the cleaved from GST ParB protein was eluted from the column, and the concentration of protein in the preparation was determined using the Bradford method. The ParB protein preparation was divided into portions of 50 μl and stored at − 80 °C.

## Western blotting

For Western blotting analyses, *S. venezuelae* cultures were set up as follows: 5 ml of liquid MYM was inoculated with spores to a starting $OD_{600} = 0.05$, and the cultures were incubated for 2–26 h at 30°C with shaking (180 rpm). Next, a 2 ml sample of the culture was collected by centrifugation (5000 × *g*, 5 min, 4 °C), washed twice with phosphate-buffered saline (PBS buffer), resuspended in 200 μl of chilled PBS buffer supplemented with Pierce™ Protease Inhibitor Tablets (Thermo Fisher Scientific, USA), and disrupted by sonication. The cell lysate was then clarified by centrifugation (12,000 × *g*, 5 min, 4 °C), and the supernatant was transferred to a fresh tube. The total protein concentration was quantified using the ROTI®Quant Universal kit (Carl Roth, Germany). Samples were prepared by mixing equal amounts of total protein (20 μg) from each lysate with 6 × SB buffer (375 mM Tris−HCl pH 6.8, 12% SDS, 0.06% bromophenol blue, 600 mM DTT, 60% glycerol), denatured at 95 °C for 10 min, and resolved by standard Laemmli acrylamide gel electrophoresis (SDS-PAGE). After electrophoresis, proteins were stained with InstantBlue Coomassie Protein Stain (Abcam, UK) (CBB, loading control) or transferred to a nitrocellulose membrane (Amersham, UK) and blocked with 2% skimmed milk (SM Gostyn, Poland) in Tris-buffered saline (TBS buffer) supplemented with 0.05% Tween-20 (TBST buffer). The blocked membrane was subsequently incubated with primary rabbit polyclonal anti-ParB antibody, mouse monoclonal anti-HaloTag antibody (Promega, USA, Cat# G921A) or mouse monoclonal anti-FLAG antibody (Sigma −Aldrich, USA, Cat# F9291-1MG). Primary antibodies were diluted 1:100 in TBST buffer, and the membrane was incubated for 1 h at room temperature, followed by washing three times with TBST buffer. Next, the membrane was incubated with secondary polyclonal antibodies: mouse anti-rabbit (Invitrogen, USA, Cat# 31464) or anti-mouse IgG (Invitrogen, USA, Cat # 62-6520) conjugated with horseradish peroxidase. The secondary antibodies were diluted 1:5000 in TBST buffer. Next, the membrane was washed as described earlier. For HRP signal detection, the membrane was washed briefly with 5 ml of SuperSignal West Pico PLUS Chemiluminescent Substrate solution (Thermo Fisher Scientific, USA), and the chemiluminescence signal was quantified using ChemiDoc XRS + (Bio-Rad, USA). The band intensities were quantified using Fiji software.

## Quantification of the *oriC/arm* ratio

To estimate the *oriC*/arm ratio, chromosomal DNA was extracted from *S. venezuelae* 5 ml cultures growing for 13–26 h or from 50 ml cultures in MYM growing for 8 to 16 h. One ml of each culture was centrifuged (1 min at 5000 rpm and room temperature). The supernatant was discarded, and the mycelium was used for the subsequent isolation of chromosomal DNA using a Genomic Mini AX *Streptomyces* kit (A&A Biotechnology) according to the manufacturer's protocol. The purified DNA was dissolved in 50 μl of DNase-free water (Invitrogen). The DNA concentration was measured at 260 nm and subsequently diluted to a final concentration of 1 ng/ml.

qPCR was performed using Power Up SYBR Green Master Mix (Applied Biosystems) with 2 ng of chromosomal DNA serving as a template for the reaction and oligonucleotides complemented to the *oriC* region (*oriC*) (gyr2_Fd, gyr2_Rv) or the right arm termini (*arm*) (arg3_Fd, arg3_Rv). The changes in the *oriC/arm* ratio were calculated using the comparative ΔΔCt method, with the arm region being set as the endogenous control. The *oriC/arm* ratio was estimated as 1 in the culture (5 ml) growing for 26 h, corresponding to the appearance of spore chains.

## Chromatin immunoprecipitation combined with next-generation sequencing (ChIP-seq)

Chromatin immunoprecipitation was conducted as described before[28,30]. Briefly, wild type, Δ*smc* and Δ*parB S. venezuelae* (WT, TM010 – three replicates and MD020, two replicates) were cultured for 5 h in 50 ml liquid MYM medium supplemented with Trace Element Solution (TES, 0.2x culture volume)[55] at 30 °C with shaking. Cultures were inoculated with 100 μl of spore suspension with $OD_{600} = 10$. After 5 h of growth, the cultures were cross-linked with 1% formaldehyde for 30 min and blocked with 125 mM glycine. Next, the cultures were washed twice with PBS buffer, and the pellet obtained from half of the culture volume was resuspended in 750 μl of lysis buffer (10 mM Tris-HCl, pH 8.0, 50 mM NaCl, 14 mg/ml lysozyme, protease inhibitor (Pierce)) and incubated at 37 °C for 1 h. Next, 200 μl of zircon beads (0.1 mm, BioSpec products) were added to the samples, which were further disrupted using a FastPrep-24 Classic Instrument (MP Biomedicals; 2 × 45 s cycles at 6 m/s speed with 5 min breaks, during which the lysates were incubated on ice). The obtained cell lysates were diluted with 750 μl IP buffer (50 mM Tris-HCl, pH 8.0, 250 mM NaCl, 0.8% Triton X-100, a Pierce™ Protease Inhibitor (Thermo Fisher Scientific, USA)), and sonicated to shear DNA into fragments ranging from 300 to 500 bp. The samples were centrifuged, and the supernatant was mixed with 2.25 μg of affinity-purified rabbit polyclonal anti-ParB antibody (the antibody was purified using *S. venezuelae* ParB immobilised on CN-Br activated Sepharose, Cytivia). Lysates were incubated with antibodies for 14 h at 4 °C, and next 80 μl of Pierce™ Protein A Magnetic BeadsG (Thermo Fisher Scientific, USA, Cat#. 88845) were added to the samples. Immunoprecipitation was performed overnight at 4 °C. Next, the magnetic beads were washed twice with IP buffer, once with IP2 buffer (50 mM Tris-HCl pH 8.0, 500 mM NaCl, 0.8% Triton X-100, Pierce™ protease inhibitors (Thermo Fisher Scientific, USA)), and once with TE buffer (10 mM Tris-HCl, pH 7.6, 10 mM EDTA), followed by resuspension in IP elution buffer (50 mM Tris-HCl pH 7.6, 10 mM EDTA, 1% SDS) and overnight incubation at 65 °C. Next, the samples were centrifuged, and proteinase K (Roche) was added to the supernatants to a final concentration of 100 μg/ml followed by incubation for 90 min at 55 °C. DNA was extracted with phenol and chloroform and subsequently precipitated overnight with ethanol. The precipitated DNA was dissolved in nuclease-free water (10 μl). The concentration of the DNA was quantified using a Qubit dsDNA HS Assay Kit (Thermo Fisher Scientific, USA).

DNA sequencing was performed by NGS Services (Switzerland) using the Illumina ChIP-Seq TruSeq protocol, which included quality control, library preparation and SE sequencing (1 × 150 bp) of amplified

fragments. The bioinformatics analysis was performed using the R package *normr* and the MACS2 programme[57,58]. The mapping of ChIP-seq data was performed using the *Bowtie2* tool (version 2.5.2)[59]. The successfully mapped reads were subsequently sorted using *samtools* (version 1.19.2)[60]. The total number of mapped reads was above $10^6$ on average. Regions specifically bound by ParB were identified using the MACS2 programme (version 2.2.9.1). Only regions with fold > 1.75 were considered for further analysis. The unspecific binding of ParB was analysed using the normr package (version 1.30.0). Reads were counted in a 100 bp long window, and the diffR function was used to find regions enriched in either wild-type or Δ*smc* strains. Only regions with FDR < 0.001 were considered to be significant.

## FLAG-SMC purification

For the FLAG-SMC pull-down assay *S. venezuelae* TM017 strain was used[28]. 1 ml of spore suspension with $OD_{600} = 0.05$ was used to inoculate 900 ml liquid MYM medium supplemented with 10 ml of TES solution, cultures were grown at 30 °C with shaking (200 rpm) for 16 h. The cells were collected by centrifugation (5000 × *g*, 10 min, 4 °C), washed twice with 50 ml of IP buffer (50 mM Tris–HCl pH 8.0, 250 mM NaCl, Triton 0.8%), supplemented with Pierce™ Protease Inhibitor Tablets (Thermo Fisher Scientific, USA) and resuspended in 2 ml of the same buffer. Next, the cells were disrupted by sonication, and the obtained cell lysates were clarified by centrifugation (9000 × *g*, 20 min, 4 °C). The clarified cell lysate was incubated overnight with 200 µl of magnetic beads coated with anti-FLAG® BioM2 antibody (ThermoFisher Scientific Cat# A36797) with constant tube rotation at 4 °C. Next, the magnetic beads were washed three times with 1 ml of IP buffer. To elute bound protein, the magnetic beads were incubated in 200 µl IP buffer (50 mM Tris–HCl pH 8.0, 250 mM NaCl, Triton 0.8%) supplemented with 3xFLAG peptide (to a final concentration of 150 µg/µl) at 4 °C overnight. Purified FLAG-SMC was stored at − 20 °C with 25% glycerol.

## CTP hydrolysis assay

Measurements of ParB protein CTPase activity were performed in Tris-HCl buffer at a final volume of 40 µl with a constant concentration of ParB protein (2 µM) and CTP (4 mM). The CTP hydrolysis was analysed in the presence of pBSK plasmid (15 µM) that contained a single *parS* sequence or scrambled *parS* and/or in the presence of FLAG-SMC (0.02 µM).

The CTP hydrolysis reaction was carried out in transparent 96-well plates for 60 min at 37 °C. The released phosphate was detected using the ATPase/GTPase Activity Assay Kit (Merck, Cat# MAK113) according to the manufacturer protocol. The standard curve prepared according to kit manufacturer guidelines was used to calculate the amount of phosphate released [µmol] in the analysed sample. The obtained absorbance results were converted into the number of µmoles of released phosphate per 1 µmol of protein [µmol/µmol of protein] in one hour. Statistical analyses were carried out using the Student's *t* test with correction for multiple testing.

## Fluorescent labelling for microscopy analyses

For the fluorescent microscopy analyses, *S. venezuelae* strains were cultured either on glass coverslips (for DNA and cell wall staining, as described before[37] or in liquid cultures in MYM medium (supplemented with Hoechst 33342 (16.,2 nM, Invitrogen, Cat# H3570) for DNA staining, NADA-green (2.,5 nM, Tocris Bio-Techne, Cat# 6648), for peptidoglycan staining) or ligands for the HaloTag protein - TMR *direct ligand* (5 nM, Promega, Cat# G299A) or Janelia Fluor 549 (0.,1 nM, Promega, Cat# GA111A). All dyes were added to the culture at the time of inoculation. Before the preparation of microscopy samples (i.e., after 5, 17, and 22 hours of growth), the cultures were washed three (for epifluorescence microscopy) to eight times (for single molecule tracking) with PBS buffer and resuspended in PBS (50 - 200 µl). Next,

the cells were spread on agarose pads prepared using gene frames (Thermo Fisher Scientific, USA) on 1.1% low-gelling temperature agarose (Sigma-Aldrich, USA, Cat# A9414-25G) in MYM or PBS, for the snapshot and FRAP analyses or single molecule tracking, respectively. Samples were covered with an 18 × 18 mm coverslip (High Precision Microscope Cover Glasses, Carl Roth, Germany).

## Time-lapse microscopy

Time-lapse fluorescence microscopy was performed using a previously described protocol[61] employing the CellAsic Onix microfluidic system using B04A plates (Merck, Germany, Cat# B04F-01-5PK). Briefly, the B04A microfluidic plate was washed with MYM medium. Next, 100 µl of spores were loaded into the microfluidic plate at a pressure of 4 psi for a duration of 2 to 10 s, depending on the density of the spore suspension. The cultures were initiated by perfusing the spores with MYM medium at a constant flow rate, maintaining a pressure of 3 psi and a temperature of 30 °C. After 3 h of pre-incubation, the "spent" MYM that had been derived from a 40 h flask culture and filter-sterilised was applied to the flow channel. For ParB-HT visualisation in the time-lapse microscopy, strains expressing *parB-HT* were cultured in the presence of the HaloTag ligand – TMR direct ligand (5 nM, Promega, Cat# G299A) and optionally in the presence of NADA-green (2.5 nM, Tocris Bio-Techne, Cat# 6648). Dyes were added to the MYM medium supporting the growth in the microfluidic system ("spent" and fresh MYM medium).

The observation was performed using a DeltaVision Elite Ultra High-Resolution microscope equipped with a CoolSnap camera and Olympus PLANApo 100x/1.40 OIL PH3 lens, and the following filter sets: EYFP-FITC (ex., 513/17 nm; em., 548/22 nm), mCherry-mCh (ex 575/25 em 625/45 nm). The images were taken every 10 min or 15 min (as indicated) for approximately 20 h with an exposure time of 50 ms at 5% transmission for the DIC channel and 80 ms at 50% transmission in the YFP and mCherry channels. The time-lapse movies were analysed using ImageJ Fiji software (version 2.14.0/1.54p)[62].

The positions of the fluorescent complexes were identified using custom protocols involving Fiji software and the R (code available at https://github.com/astrzalka/findpeaks). For statistical analysis two-sided Student's *t* test or Wilcoxon test, with an adjustment for multiple testing if necessary, were used.

## Fluorescence recovery after photobleaching

For FRAP experiments the samples on agarose were prepared from 5 h liquid cultures in 2 ml MYM medium supplemented with Janelia Fluor-549 HaloTag ligand (0.1 nM, Promega, Cat# GA111A, added to the culture at the time of inoculation). Cultures were washed two times with 2 ml of PBS, then resuspended in PBS and loaded on microscopy slides covered with agarose pads (1% low-gelling temperature agarose in PBS). FRAP analyses were carried out using Stellaris 8 confocal microscope (Leica GmbH, Germany), equipped with HC PL APO CS2 100x/1.40 OIL lens and at a temperature set to 30 °C. 540 nm laser line, originating from WLL2 (with a total power >1.8 mW), set at 80% of the total power, was used to image and bleach the sample. Fluorescent complexes located in proximity to the hyphal tip were selected to bleach. Before bleaching, 3 frames with no interval were acquired with a laser power set to 2.32%. Then the rectangular area covering the fluorescent complex was bleached by acquiring 3 consecutive frames with a laser power set to 95%. In the post-bleach phase, 19 frames were acquired with an interval set to 30 seconds (total post-bleach acquisition time was 9.5 min) and a laser power set to 2.32%. Image analysis was performed in Fiji using the Stowers set of plug-ins to get the FRAP curves as well as the recovery times and the sizes of mobile fractions[62] and R packages. In the FRAP experiment, each frame was averaged two times (with the exemption of the bleaching phase) and had a size of 168 × 75 pixels. The pixel size was set to 70 nm, whereas the pixel dwell time was set to 3.2.

## Single molecule tracking (SMT)

The microscopy samples for SMT were prepared from 5, 17 and 22 h liquid cultures in 2 ml MYM medium supplemented with Janelia Fluor-549 HaloTag ligand (0,1 nM, Promega, Cat# GA111A), as described above, but the cultures were washed 7 times with 2 ml PBS before slide preparation. SMT analyses were carried out using the Zeiss Elyra 7 microscope equipped with Alpha Plan-APO 100x/1.46 Oil DIC VIS lens, sCMOS + emCCD and Andor EM-CCD cameras, and following laser lines: 405 (50 mW), 488 (100 mW), 561 (500 mW), 633 nm (500 mW). The imaging was performed with a 561 nm laser with a fixed power range of 80 %. From 10,000 to 20,000 frames were collected with an exposure time of 20 ms. The experiment was conducted at a constant temperature of 30 °C. Fiji plugin Trackmate (version 7.14.0)[63], with a maximum linking distance of 0.5 µm, was used for single molecule tracking. Only tracks with at least 4 steps were considered for further analysis. Cell outlines were prepared using the Oufti programme[64]. SMTracker (version 2.0) was used to analyse ParB-HT mobility in hyphae[65]. Square Displacement analysis (SQD) was used to divide the analysed ParB-HT molecule tracks into two diffusive populations based on their apparent diffusion constant (D).

## Reporting summary

Further information on research design is available in the Nature Portfolio Reporting Summary linked to this article.

## Data availability

The raw ChIP-Seq data, as well as the processed data generated in this study (shown in Fig. 6), have been deposited in the ArrayExpress database (EMBL-EBI) under accession code E-MTAB-14547. Images and Movies are available in BioImage Archive: Fig. 1 and Supplementary Fig. S2 under accession code S-BIAD1808, Fig. 2 and Supplementary Fig. S4 under accession code S-BIAD1807, Fig. 3 and Supplementary Fig. S8 under accession code S-BIAD1811, Supplementary Fig. S5 under accession code S-BIAD1813, Supplementary Fig. S3 under accession code S-BIAD1814 or in Figshare: Fig. 4 data https://doi.org/10.6084/m9.figshare.28730597, Supplementary Fig. S7 data https://doi.org/10.6084/m9.figshare.28730699, Supplementary Fig. S1D data: https://doi.org/10.6084/m9.figshare.28730795. Source data are provided in this paper.

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

## Acknowledgements

This work was funded by the Polish National Science Centre: OPUS grant 2018/31/B/NZ1/00614 (to D.J.). We thank Tung Le (John Innes Centre, Norwich,UK) for the discussion of the results and helpful comments and Gemma Alderton (www.biosciedit.co.uk) for manuscript corrections. We thank Marcin Wolański (Dept. of Molecular Microbiology, Univeristy of Wroclaw) for the help with BioRender.com.

## Author contributions

Conceptualisation and experiment design (D.J.), methodology including strains construction and protein purification (K.P., J.D.-K.), microscopy and data analysis (K.P., A.S. and M.M.); ChIP-Seq libraries preparation and data analysis (K.P. and A.S.); figures preparation (K.P., A.S. and D.J.); writing the manuscript (D.J.); manuscript revisions (K.P., A.S., M.Sz. and D.J.).

## Competing interests

The authors declare no competing interests.
