## [Peer Review file · Nature Communications]

SMC modulates ParB engagement in segregation complexes in *Streptomyces*

Corresponding Author: Professor Dagmara Jakimowicz

Version 0:

Reviewer comments:

Reviewer #1

(Remarks to the Author)

This study examines the interplay between bacterial ParB and SMC proteins, specifically the influence of the latter on the former, primarily during sporulation in *Streptomyces*. The ParB of the ParABS partition systems binds specifically to multiple parS centromere-like sites that are generally clustered in the origin region of bacterial chromosomes. ParB binds to parS and then slides away or spreads to adjacent DNA in a CTP-dependent way, forming large segregation complexes. ParB also recruits SMC proteins to the origin region, and multiple studies have examined this ParB recruitment of SMC, but here the authors investigate the influence of SMC on ParB activities. This interplay between SMC and ParB has not been previously explored. SMC proteins have both condensing and cohesin roles in bacteria, and are essential for proper chromosome conformation and arrangements during the bacterial cell cycle.

The study uses primarily a cell biology approach, and examines the formation and duration of ParB “segrosomes” during spore development, with and without SMC. Without SMC, they find that both the time of onset of assembly, and duration of ParB segrosomes are shortened, arguing that SMC stabilizes ParB segrosomes. Using single-molecule tracking they find that normally ParB complex disassembly correlates with increased ParB mobility in cells (interpreted as free, not DNA bound, ParB), and that this mobility is reduced or delayed in cells without SMC. Taken together with other experiments they suggest that without SMC, ParB complexes disassemble faster but they remain bound to DNA non-specifically elsewhere, but the nature of this binding (eg direct vs indirect via ParA for example) is not known. Finally, they show that SMC inhibits the CTPase activity of ParB, providing an explanation why it stabilizes ParB complexes because CTP hydrolysis promotes ParB unclamping from DNA. The study is quite thorough and there are a lot of careful and sophisticated fluorescence and other approaches described here, and the authors carefully check for potential changes in protein expression or activity.

1. The main outstanding question is whether the effect of SMC is direct (via protein-protein interaction) or indirect (via chromosome conformation/condensation/cohesion), or both, and the picture is incomplete without testing this distinction. It is clear from other studies that there is a ParB-SMC protein-protein interaction. However SMC has a major effect on chromosome conformation, and higher-order ParB complex assembly is proposed to involve DNA bridging interactions, so it is formally possible that the SMC effects seen here are because SMC provides the proper DNA arrangement to form and stabilize these ParB complexes. The CTPase experiments here include plasmid DNA, so they do not exclude the possibility that the SMC effect is mediated via DNA (but see comment #2). Given that ParB also recruits SMC *in vivo*, it may be difficult to separate direct and indirect effects mutationally *in vivo*. However, since the region of ParB that interacts with SMC is mapped and mutated in *B. subtilis*, it seems that such a *Streptomyces* mutant would confirm whether this interaction is necessary using the CTPase assay.

2. The CTPase assays do not include ATP, so it is possible that the authors conclude that SMC is not binding to DNA (which partially answers comment #1). However it should be directly demonstrated, for example by including ATP in the assay. This would require a different assay since here they measure phosphate release, but the use of radiolabelled CTP should allow the discrimination between CTPase and ATPase activities. In any case, to address both comments 1 and 2, the authors should discuss the potential role of DNA condensation and arrangement on ParB complex stabilization.

3. The proposal that SMC stabilizes the clamp form of ParB could be tested using the cross-linking approach that has been reported by other labs.

Minor:

4. Fig S6, showing similar ParB levels by Western. It would be nice to have a control showing that the experiment was in a linear (or close) range of detection; since all the bands examined were of equal intensity. For example, they could show that the intensity varies when half or double the amount of protein is run on the gel.

Reviewer #2

(Remarks to the Author)

It has long been established that ParB and SMC proteins are crucial components in bacterial chromosome organization and segregation. They are part of the ParABS system, which is essential for the accurate partitioning of chromosomes during cell division. Moreover, it is also demonstrated that ParB and SMC form a complex involving their CTP-binding and joint domains, respectively.

In the present study, the authors investigate the role of ParB and SMC during chromosome organization and segregation of *Streptomyces venezuelae*. In essence, they confirm that SMC modulates ParB activity *in vivo*. As a sort of novel finding, which could have been assumed from previous data, absence of SMC reduces the spreading of ParB. *In vitro*, they clearly show that SMC somehow restricts the CTPase activity of ParB. From that data, they essentially create a model in which SMC creates a positive feedback on the ParB nucleoprotein complex.

The work is experimentally sound, and we have no major criticism on the experiments. The manuscript is well written. Our main point is that the study represents an incremental step in our understanding of the ParABS system, with the "positive feedback" staying widely hypothetical.

This impression could be resolved by a more mechanistic explanation of how exactly SMC would restrict the ParB CTPase. -E.g., Which step of the CTPase cycle is inhibited? (i.e., release of CDP, uptake of CTP, hydrolysis ...). Measuring kinetics of ParB in the absence and presence of SMC might already provide insights.

-What happens with an inactive CTPase variant *in vivo*? How would this variant compare to an SMC knockout, or an SMC variant unable to interact with ParB (compare with: PMID: 36044845).

Minor points:

-Please describe better in the Materials&Methods section: How was ParB purified for the *in vitro* experiments? Description of the fluorescent labelling for microscopic analyses was not clearly described. Would be great if that could be checked for clarity again.

Reviewer #3

(Remarks to the Author)

Reviewer #4

(Remarks to the Author)

In this paper, Pawlikiewicz et al. report an effect of SMC on the dynamics of ParB in *Streptomyces venezuelae*. Using a functional ParB-HaloTag (-HT) fusion, the authors show that segrosomes became equally spaced at the onset of sporulation and then disassembled at later stages of sporogenic development, concomitant with the disassembly of FtsZ rings. In the Δsmc background, by contrast, the equipartitioning of segrosomes and, in particular, their disassembly occur at a significantly earlier time point, while the duration of the sporulation process is slightly longer. Consistent with these observations, single-molecule tracking analysis shows that the fraction of mobile molecules increases over the course of sporogenic development, suggesting an increase in the number of ParB molecules that are not included in segrosomes. However, while the absence of SMC does not affect the mobility of ParB at the onset of sporulation, it leads to a considerable increase in the fraction of immobile molecules at later stages of sporogenic development. A similar increase in the immobile fraction is observed in vegetative Δsmc cells. Nevertheless, FRAP analysis suggests that the apical segrosome complex shows higher turnover under these conditions. Using ChIP-seq analysis, the authors find that ParB shows lower association with parS sites and apparently less spreading in the Δsmc background. Finally, the authors show that SMC decreases the CTPase activity of ParB to ~50% *in vitro*. Based on these results, they suggest that SMC increases the stability of segrosomes by keeping ParB in the clamped state and thus promoting its spreading activity.

Overall, the authors demonstrate that the absence of SMC affects the long-term stability of segrosomes over the course of sporogenic development and the short-term assembly dynamics of segrosome complexes. However, although these findings are interesting, the data are still somewhat preliminary and not sufficient to support the conclusions drawn. Specifically, the following issues should be addressed before publication of the paper:

Major issues:

1. The authors observe that segrosomes disassemble at an earlier time point during sporogenic development in the Δsmc

mutant and explain this behavior with the effect of SMC on the assembly dynamics of segrosomes. However, the turnover rates of segrosomes, as determined by FRAP analysis, are in the range of 2-4 min, whereas the interval over which segrosomes gradually disassemble during sporogenic development is much longer. Thus, the stability of segrosomes at the long time scale of the developmental program cannot be determined by their short-term assembly dynamics. Elevated ParB dissociation rates would only lead to smaller segrosome sizes throughout the lifetime of developmental program. The results obtained rather suggest that there are so-far unknown, developmentally regulated factors governing segrosome assembly/disassembly, e.g. by blocking access to the parS sites. This notion is also supported by the observation that the levels of SMC and ParB do not change of the course of development, which suggests that their interaction must be modulated as the developmental program proceeds. Along these lines, the authors observed that the lack of SMC has no effect on the dynamics of segrosomes in early sporogenic cells. If the effect of SMC on ParB dynamics were only exerted through its direct interaction with ParB, it should be observed in all growth and developmental phases. It is possible that the accumulation or function of a putative regulatory factor is affected by the absence of SMC, explaining the changes in the time of segrosome appearance and disappearance.

2. The FRAP analysis focused on the apical segrosome complex, which differs from other segrosomes by its interaction with the TIPOC. Could this interaction affect the dynamics of the complex? The FRAP experiment measures the recovery of fluorescence in the apical segrosome. However, the ParB-HT molecules in this complex were bleached, so that the recovery of fluorescence depends on the pool of freely diffusible proteins or the dissociation rates of other segrosome complexes. The rates observed do therefore not necessarily reflect the turnover of the apical complex. Could a faster recovery rate in Δsmc cells explained by the fact that less ParB-HT molecules are stably associated with segrosome complexes and more of them are non-specifically bound to the nucleoid, so that they have a higher dissociation rate from their low-affinity binding sites and are thus more readily available to associate with the bleached apical segrosome?

3. The authors state that ParB spreading is diminished in the Δsmc background. Could the lower signal in the surroundings of the parS sites (which indicate spreading) simply be explained by the generally lower number of ParB-HT molecules that are associated with parS sites in this condition? It looks like ratio of reads for the main peak (0 bp) and the tails (around \pm 500 bp) (Figure 4A) are quite similar for the wild type and Δsmc mutant, suggesting that the overall shapes of the distributions are not dramatically different.

4. The authors do not take into account the possibility that the lower occupancy of parS sites in the Δsmc mutant could result from defects in chromosome organization that may affect the rate of ParB clamp loading or the postulated bridging activity of ParB.

5. The effect of SMC on the CTPase activity of ParB is relatively moderate. Moreover, the number of SMC molecules per cell is very low (~80 in *B. subtilis*) and these molecules are distributed across the entire chromosome. ParB, on the other hand, is typically significantly more abundant and condensed in segrosome complexes. Therefore, most ParB molecules would likely remain unaffected by SMC. It is therefore not clear how a direct interaction between SMC and ParB can determine the global dynamics of segrosomes, as proposed by the authors in their final model.

Other issues:

6. The introduction is, in part, unstructured and does not provide the general reader with a clear explanation of the background knowledge required to understand the topic. The authors should introduce the formation of segrosomes (with the proper literature cited), the mode of action of ParABS systems (including a brief description of the segregation mechanism), the composition and role of SMC proteins, and the previous knowledge about the interplay between ParA, ParB and SMC before explaining the specific functions of these proteins in *Streptomyces* species.

7. It appears that the different strains shown in Figure S1A accumulate ParB-HT to quite different levels. Why is the level in the absence of aTet higher than in the presence of the inducer? The Western blot analysis performed with an anti-ParB antibody should include the wild-type strain without any deletions or tags. Otherwise, it is difficult to assess whether the levels of the tagged proteins are indeed similar to that in the wild-type strain. Please indicate the antibodies used in the Figure.

8. The Western blots in Fig. S6A show that there are not marked changes in the levels of ParB-HT over the course of sporogenic development in the wild-type and Δsmc backgrounds. However, since the two strains are analyzed on two different gels, it is not possible to conclude from these results that the lack of SMC does not change the ParB-HT levels.

9. How do the authors explain the observation that, in vegetative Δsmc cells, the immobile ParB-HT molecules are not those forming the apical segrosome complex? Are the proteins no longer clamped onto the DNA in this complex?

10. Faster segrosome disassembly in the absence of SMC should lead to fainter segrosome foci. The authors should measure the average intensity of segrosome complexes in the wild-type and Δsmc mutant to test this prediction.

Comments/suggestions:

- Line 39: "Turnover of segrosomes" or "assembly and disassembly kinetics of ParB"
- Line 43: "several" instead of "numerous"
- Line 52: "bacterium"
- Lines 72/73: Perhaps "in the model species *S. venezuelae*, which allows ..."

- Line 96: "were followed"
- Line 97ff: "fluorescence microscopy"
- Line 101: "diffused" is not the correct word here. Perhaps "gradually disassembled and completely disappeared"
- Line 134: Perhaps clearer: "the more rapid disassembly"
- Figure 3A: The x-axes should be labeled in the graphs shown.
- Figure 3B: Did the authors use a global definition of the diffusion constants in the different strains or conditions? If so, this should be indicated.
- Figure 4: The proper term for Tau is "recovery half-time"
- Throughout: The diffusion constant is usually represented by a capital "D".
- Lines 137ff: The authors frequently refer to the diffusion constant, which averages the mobility of all molecules analyzed) or step sizes when reporting differences in protein mobility. The description of their results would be much clearer if they stated at the beginning that the diffusion behavior of ParB-HT suggests the existence of two distinct populations: an immobile population that is likely associated with DNA in segrosomes, and a mobile population that is likely freely diffusing. Then, the arguments should be made based on changes in the fraction sizes of these two populations.
- Line 181: "MSD" is the abbreviation of "mean square displacement"
- Figure 3F: Please show the cell outlines.
- Line 224: This has also been shown for other ParB homologs.
- Line 307: "corroborates the results of earlier studies"
- Line 312ff: The segrosomes are still equally spaced in the Δ smc mutant (and even earlier than in the wild type), suggesting that chromosome segregation is not affected.
- The references should be properly formatted and checked for errors and missing information.
- Methods: How were the Western blots performed? Did the authors use equal volumes of culture or normalize to OD when preparing the samples? This makes a difference when assessing changes in the concentration of proteins over time.

Version 1:

Reviewer comments:

Reviewer #1

(Remarks to the Author)

The authors have provided new experiments and discussion to address many of the reviewers' comments. Notably from my perspective, they examined whether direct ParB:SMC interactions were necessary using ParB mutants with diminished SMC interactions, and the effect, although mild, does support the idea that direct interaction plays a role in CTPase activity by ParB. They admit that indirect effects via SMC roles in chromosome conformation may also be important, which does seem likely to me. These revisions and their response to my comments have been addressed to my satisfaction.

Reviewer #2

(Remarks to the Author)

The authors have addressed all points we have raised.

Reviewer #4

(Remarks to the Author)

In the revised version of the manuscript, the authors have performed several additional experiments to support their conclusions. Most importantly, they have generated a variant of ParB with amino acid exchanges in the putative SMC binding site to verify that the SMC-mediated decrease in the CTPase activity of ParB depends on a direct interaction between the two proteins. Although the additional data further support the hypothesis that SMC in some way affects segrosome assembly, there are still some issues that should be addressed:

1. The new data provided in Figure S10 do not address the previous question of whether the overall shapes of the ParB distributions obtained for WT and Δ smc cells at individual parS sites are similar. A comparison of distributions measured at different parS sites is not valid, because the sliding behavior of ParB is affected by different factors such as the presence roadblocks. If the WT (red) and Δ smc (blue) distributions in panels A are normalized so that their maxima at position 0 bp are identical, then the shapes and the width of the distributions are comparable. The same is true for the ChIP-seq peaks at other parS sites. This suggests that ParB is not less stably associated with the centromeric region after release from parS but rather that less ParB is loaded at parS in the first place. This would mean that SMC does not affect complex stability and spreading (as suggested in the abstract and Discussion) but rather the loading step.

2. The authors now show that SMC needs to directly interact with ParB to affect its CTPase activity in vitro. However, I still find it difficult to imagine how a small number of SMC complexes distributed throughout the chromosomal DNA (and thus spatially separated from ParB) are able continuously affect the stability of several hundred ParB clamps sliding in the centromeric regions by means of direct protein-protein interactions (as suggested as the most likely mechanism in the Discussion). It appears more likely that SMC affects the behavior of ParB specifically at the loading stage, thereby

modulating the overall number of ParB clamps associated with the centromeric region. Consistent with this notion, the observation that SMC affects the ParB CTPase activity at substoichiometric concentrations suggests a catalytic effect rather than a stable interaction. Would it be possible that SMC is loaded by closed ParB clamps and SMC thus promotes the transition of ParB to the closed state at parS? It is difficult to tell whether the relatively small decrease in the CTPase activity observed in vitro is really physiologically relevant and not caused by the free diffusibility of ParB and SMC in solution, which enables frequent contacts throughout the CTPase cycle (in contrast to the in vivo situation). Generally, it is difficult to draw firm conclusions from this result without knowing whether/how the amino acid exchanges in the putative SMC binding site affect the overall CTPase cycle of ParB or the functionality of ParB in vivo.

3. The authors relate the fast recovery of ParB complexes to the higher levels of ParB dimers that are non-specifically associated to chromosomal DNA. At the same time, they argue that this fast recovery indicates a shortened life-time of ParB complexes. Would it be possible that the lack of SMC leads to less ParB loading at parS and thus to a higher level of non-specifically associated ParB, which may be less stably bound and thus, in turn, lead to faster recovery?

4. How do the authors explain the fact that the intensity of ParB foci does not differ between WT and Δ smc cells?

5. I still think that it is not justified to draw a direct connection between differences in the short-term kinetics of segrosome assembly and the long-term stability of these complexes over the course of the developmental cycle. Segrosomes turn over many times during this cycle, so that changes in the equilibrium level of ParB within segrosomes should be dependent on factors other than the presence of the SMC-ParB interaction (even though lower equilibrium levels of ParB in the segrosomes may make such additional regulatory effects more easily detectable).

- Line 304: "CTP binding"

Version 2:

Reviewer comments:

Reviewer #4

(Remarks to the Author)

In the revised version of their manuscript, the authors present an alternative model that better aligns with their experimental data. Overall, the manuscript now offers interesting insights into the interplay between SMC and ParB in *S. venezuelae*, and I am looking forward to seeing this paper in print.

We are grateful to the Reviewers for their thoughtful and helpful feedback. Below, we provide our point-by-point responses to each comment.

REVIEWER COMMENTS

Reviewer #1 (Remarks to the Author):

This study examines the interplay between bacterial ParB and SMC proteins, specifically the influence of the latter on the former, primarily during sporulation in *Streptomyces*. The ParB of the ParABS partition systems binds specifically to multiple parS centromere-like sites that are generally clustered in the origin region of bacterial chromosomes. ParB binds to parS and then slides away or spreads to adjacent DNA in a CTP-dependent way, forming large segregation complexes. ParB also recruits SMC proteins to the origin region, and multiple studies have examined this ParB recruitment of SMC, but here the authors investigate the influence of SMC on ParB activities. This interplay between SMC and ParB has not been previously explored. SMC proteins have both condensing and cohesin roles in bacteria, and are essential for proper chromosome conformation and arrangements during the bacterial cell cycle.

The study uses primarily a cell biology approach, and examines the formation and duration of ParB “segrosomes” during spore development, with and without SMC. Without SMC, they find that both the time of onset of assembly, and duration of ParB segrosomes are shortened, arguing that SMC stabilizes ParB segrosomes. Using single-molecule tracking they find that normally ParB complex disassembly correlates with increased ParB mobility in cells (interpreted as free, not DNA bound, ParB), and that this mobility is reduced or delayed in cells without SMC. Taken together with other experiments they suggest that without SMC, ParB complexes disassemble faster but they remain bound to DNA non-specifically elsewhere, but the nature of this binding (eg direct vs indirect via ParA for example) is not known. Finally, they show that SMC inhibits the CTPase activity of ParB, providing an explanation why it stabilizes ParB complexes because CTP hydrolysis promotes ParB unclamping from DNA. The study is quite thorough and there are a lot of careful and sophisticated fluorescence and other approaches described here, and the authors carefully check for potential changes in protein expression or activity.

1. The main outstanding question is whether the effect of SMC is direct (via protein-protein interaction) or indirect (via chromosome conformation/condensation/cohesion), or both, and the picture is incomplete without testing this distinction. It is clear from other studies that there is a ParB-SMC protein-protein interaction. However SMC has a major effect on chromosome conformation, and higher-order ParB complex assembly is proposed to involve DNA bridging interactions, so it is formally possible that the SMC effects seen here are because SMC provides the proper DNA arrangement to form and stabilize these ParB complexes. The CTPase experiments

here include plasmid DNA, so they do not exclude the possibility that the SMC effect is mediated via DNA (but see comment #2). Given that ParB also recruits SMC *in vivo*, it may be difficult to separate direct and indirect effects mutationally *in vivo*. However, since the region of ParB that interacts with SMC is mapped and mutated in *B. subtilis*, it seems that such a *Streptomyces* mutant would confirm whether this interaction is necessary using the CTPase assay.

We appreciate the reviewer's comment, as constructing and testing a ParB mutant that does not interact with SMC reinforces the notion that direct ParB–SMC interactions are essential for the observed stabilization of ParB on DNA.

Based on studies in *B. subtilis*, we generated such *S. venezuelae* ParB variant by substituting three amino acids in the non-conserved loop (TDR132-134 to AAT, equivalent to L69S, K70E, named 3A region in Bock et al. 2022l) and introducing the D164R mutation (equivalent to E101K). The ParB_{SMC-} variant was purified and tested in a CTPase activity assay. While its CTPase activity was reduced, likely due to the introduced modifications being close to the CTP-binding motif (GERR 140-143), it remained dependent on DNA with the *parS* sequence, confirming its interaction with DNA.

Notably, in the presence of FLAG-SMC, the CTPase activity of ParB_{SMC-} variant was only marginally affected. The impact of SMC was significantly weaker than that observed for wild-type ParB, confirming that direct ParB–SMC interactions are necessary for SMC's effect on ParB's CTPase activity, spreading, and complex stability.

We have now included these additional data in **Figure 6** and modified the **Results section** (Lines 258-278*) to include a description of this protein variant. Please note that in this repeated assay, we observed higher levels of phosphate release for wild-type ParB, most likely due to differences in protein or plasmid DNA preparations. Although the absolute CTPase activity of wild-type ParB appears is higher in this dataset, the inhibitory effect of SMC and the dependence on *parS* remain consistent with our initial results. To ensure comparability, we are replacing the entire dataset with results obtained using the new preparations. We have also added the comment on ParB_{SMC-} variant in **Discussion** (lines 293-294, 322-325*).

2. The CTPase assays do not include ATP, so it is possible that the authors conclude that SMC is not binding to DNA (which partially answers comment #1). However it should be directly demonstrated, for example by including ATP in the assay. This would require a different assay since here they measure phosphate release, but the used of radiolabelled CTP should allow the discrimination between CTPase and ATPase activities. In any case, to address both comments 1 and 2, the authors should discuss the potential role of DNA condensation and arrangement on ParB complex stabilization.

Indeed, our assay did not specifically test for SMC binding to DNA in the presence of ATP. However, it is important to note that FLAG-SMC was pulled down from *S. venezuelae* cell lysate, and it could likely co-purify with DNA. Attempts to include DNase treatment or conduct more complex nucleotide hydrolysis assays proved challenging and did not yield conclusive results.

Based on the data obtained and the application of the ParB_{SMC} variant, we infer that direct ParB-SMC interactions are required for inhibition of ParB CTPase activity by SMC. However, we cannot conclude

that this interaction alone is sufficient, nor can we exclude the possibility that SMC binding to DNA contributes to its impact on ParB complex.

3. The proposal that SMC stabilizes the clamp form of ParB could be tested using the cross-linking approach that has been reported by other labs.

We agree with the reviewer that extensive structural analyses would provide further insights into the mechanism of ParB and SMC cooperation. We considered using HDX experiments to examine the impact of SMC on ParB conformation; however, due to limitations in SMC purification and extensive optimization required either for HDX or crosslinking experiments, we were unable to perform these experiments. Nevertheless, further structural analyses of *S. venezuelae* ParB are part of an ongoing project in our department.

Minor:

4. Fig S6, showing similar ParB levels by Western. It would be nice to have a control showing that the experiment was in a linear (or close) range of detection; since all the bands examined were of equal intensity. For example, they could show that the intensity varies when half or double the amount of protein is run on the gel.

In the cell lysate analysis by SDS-PAGE (shown in FigS1 and in Fig S6) always the same amount of total protein was loaded. Figure S1 demonstrates that ParB levels fall within the linear range of detection.

Reviewer #2 (Remarks to the Author):

It has long been established that ParB and SMC proteins are crucial components in bacterial chromosome organization and segregation. They are part of the ParABS system, which is essential for the accurate partitioning of chromosomes during cell division. Moreover, it is also demonstrated that ParB and SMC form a complex involving their CTP-binding and joint domains, respectively.

In the present study, the authors investigate the role of ParB and SMC during chromosome organization and segregation of *Streptomyces venezuelae*. In essence, they confirm that SMC modulates ParB activity in vivo. As a sort of novel finding, which could have been assumed from previous data, absence of SMC reduces the spreading of ParB. In vitro, they clearly show that SMC somehow restricts the CTPase activity of ParB. From that data, they essentially create a model in which SMC creates a positive feedback on the ParB nucleoprotein complex.

The work is experimentally sound, and we have no major criticism on the experiments. The manuscript is well written. Our main point is that the study represents an incremental step in our understanding of the ParABS system, with the “positive feedback” staying widely hypothetical.

This impression could be resolved by a more mechanistic explanation of how exactly SMC would restrict the ParB CTPase.

-E.g., Which step of the CTPase cycle is inhibited? (i.e., release of CDP, uptake of CTP, hydrolysis ...). Measuring kinetics of ParB in the absence and presence of SMC might already provide insights.

Based on the data obtained so far, we conclude that the inhibition of CTPase activity by SMC does not result from impaired CTP loading. This conclusion is supported by the observation that CTP loading promotes ParB binding to *parS* and spreading (Szymczak et al. <https://doi.org/10.1101/2024.12.04.625172>), while ChIP-seq analyses show enhanced ParB binding to *parS* in the presence of SMC (wild type strain) as compared to the strain lacking SMC, suggesting that CTP binding is not diminished. We have now expanded the **Discussion** section to include this conclusion (lines 301-3015*).

However, we are unable to determine whether SMC interferes with nucleotide hydrolysis or its release by ParB. Addressing this question would require extensive structural analyses. As mentioned in our response to Comment 3 of Reviewer 1, we recognize that structural studies would provide further insights into the ParB-SMC interaction. We had considered using HDX experiments to examine the impact of SMC on ParB conformation; however, due to limitations in SMC purification, and required extensive optimization of these assays, we were unable to conduct these experiments. Nevertheless, further structural analyses of *S. venezuelae* ParB are part of an ongoing project in our department.

-What happens with an inactive CTPase variant in vivo? How would this variant compare to an SMC knockout, or an SMC variant unable to interact with ParB (compare with: PMID: 36044845). -

The *S. coelicolor* ParB variants with abolished CTPase activity were extensively studied in vitro and in vivo, and these results are described in the manuscript currently in revision (Szymczak J. et al. <https://doi.org/10.1101/2024.12.04.625172>). The data presented in that work clearly show that CTPase activity is required for efficient chromosome segregation and segrosome formation during *Streptomyces* sporogenic development.

Minor points:

-Please describe better in the Materials&Methods section: How was ParB purified for the in vitro experiments? Description of the fluorescent labelling for microscopic analyses was not clearly described. Would be great if that could be checked for clarity again.

We have expanded the description of the labelling for microscopy analyses, and we have added information on ParB and ParB_{SMC}- purification to the Supplementary information.

Reviewer #3 (Remarks to the Author):

Reviewer #4 (Remarks to the Author):

In this paper, Pawlikiewicz et al. report an effect of SMC on the dynamics of ParB in *Streptomyces venezuelae*. Using a functional ParB-HaloTag (-HT) fusion, the authors show that segrosomes became equally spaced at the onset of sporulation and then disassembled at later stages of sporogenic development, concomitant with the disassembly of FtsZ rings. In the Δsmc background, by contrast, the equipartitioning of segrosomes and, in particular, their disassembly occur at a significantly earlier time point, while the duration of the sporulation process is slight longer. Consistent with these observations, single-molecule tracking analysis shows that the fraction of mobile molecules increases over the course of sporogenic development, suggesting an increase in the number of ParB molecules that are not included in segrosomes. However, while the absence of SMC does not affect the mobility of ParB at the onset of sporulation, it leads to a considerable increase in the fraction of immobile molecules at later stages of sporogenic development. A similar increase in the immobile fraction is observed in vegetative Δsmc cells. Nevertheless, FRAP analysis suggests that the apical segrosome complex shows higher turnover under these conditions. Using ChIP-seq analysis, the authors find that ParB shows lower association with parS sites and apparently less spreading in the Δsmc background. Finally, the authors show that SMC decreases the CTPase activity of ParB to ~50% *in vitro*. Based on these results, they suggest that SMC increases the stability of segrosomes by keeping ParB in the clamped state and thus promoting its spreading activity.

Overall, the authors demonstrate that the absence of SMC affects the long-term stability of segrosomes over the course of sporogenic development and the short-term assembly dynamics of segrosome complexes. However, although these findings are interesting, the data are still somewhat preliminary and not sufficient to support the conclusions drawn. Specifically, the following issues should be addressed before publication of the paper:

Major issues:

1. The authors observe that segrosomes disassemble at an earlier time point during sporogenic development in the Δsmc mutant and explain this behavior with the effect of SMC on the assembly dynamics of segrosomes. However, the turnover rates of segrosomes, as determined by FRAP analysis, are in the range of 2-4 min, whereas the interval over which segrosomes gradually disassemble during sporogenic development is much longer. Thus, the stability of segrosomes at the long time scale of the developmental program cannot be determined by their short-term assembly dynamics. Elevated ParB dissociation rates would only lead to smaller segrosome sizes throughout the lifetime of developmental program. The results obtained rather suggest that there are so-far unknown, developmentally regulated factors governing segrosome assembly/disassembly, e.g. by blocking access to the parS sites. This notion is also supported by the observation that the levels of SMC and ParB do not change of the course of development, which suggests that their interaction must be modulated as the developmental program proceeds. Along these lines, the authors observed that the lack of SMC has no effect on the dynamics of segrosomes in early sporogenic cells. If the effect of SMC on ParB dynamics were only exerted through its direct interaction with ParB, it should be observed in all growth and developmental phases. It is possible that the accumulation or function of a putative regulatory factor is affected by

the absence of SMC, explaining the changes in the time of segrosome appearance and disappearance.

In response to the comment regarding the effect of SMC on ParB dynamics that should be observed in all growth and developmental phases, we note that the effect of the SMC elimination is observed not only in sporogenic cells but also during vegetative growth (FRAP, ChiP seq and SMT experiments). We agree that our model system provides some challenges due to *Streptomyces* complex development and the presence of multiple chromosomes in the cell, but on the other hand, it offers a unique opportunity to study the disassembly of PaB complexes in sporogenic cells (we now added this comment in the **Discussion**, lines 346-348*). We acknowledge that time-lapse microscopy and FRAP analyze ParB complex dynamics at different time scales. However, the observed higher turnover rate of ParB-HT in the absence of SMC, as seen in FRAP, aligns with the less efficient spreading of ParB detected by CHIP-seq, both corresponding to lowered complex stability in a longer time scale. While we suggest that less stable complexes may disassemble more efficiently or fail to reassemble as effectively as in wild-type conditions, we agree that other developmental factors may also influence ParB complex stability. Given the previously established role of ParA in the assembly of ParB complexes (as shown by Donczew et al. 2016), we infer that the fate of ParB complexes and the release of ParB proteins from these complexes depend significantly on ParA. However, we cannot rule out the possibility that other developmental factors may also play a role in ParB dynamics. We have now modified **the Discussion** to include this comment (lines 359-365*).

2. The FRAP analysis focused on the apical segrosome complex, which differs from other segrosomes by its interaction with the TIPOC. Could this interaction affect the dynamics of the complex? The FRAP experiment measures the recovery of fluorescence in the apical segrosome. However, the ParB-HT molecules in this complex were bleached, so that the recovery of fluorescence depends on the pool of freely diffusible proteins or the dissociation rates of other segrosome complexes. The rates observed therefore do not necessarily reflect the turnover of the apical complex. Could a faster recovery rate in Δsmc cells explained by the fact that fewer ParB-HT molecules are stably associated with segrosome complexes and more of them are non-specifically bound to the nucleoid, so that they have a higher dissociation rate from their low-affinity binding sites and are thus more readily available to associate with the bleached apical segrosome?

Indeed, we believe that the fast recovery is explained by a higher fraction of ParB molecules that are not involved in complex formation (as shown in **Fig 3E**). We have now added this explanation in the Discussion (lines 290-292*). To address the comment on the difference of the apical ParB complex and the other ParB complex in vegetative hyphae, we now include the comparison of ParB -HT mobility in the tip proximal and the tip distal region (**Fig. S8B**). While the fraction of immobile ParB molecules is somewhat lower in the tip-distal region and the impact of elimination of SMC is slightly lower, still the difference between ParB mobility in the presence and the absence of SMC is significant.

3. The authors state that ParB spreading is diminished in the Δsmc background. Could the lower signal in the surroundings of the parS sites (which indicate spreading) simply be explained by the generally lower number of ParB-HT molecules that are associated with parS sites in this condition? It looks like ratio of reads for the main peak (0 bp) and the tails (around ± 500 bp) (Figure 4A) are quite

similar for the wild type and Δsmc mutant, suggesting that the overall shapes of the distributions are not dramatically different.

We have checked this possibility by comparing the binding profile for two single *parS* sites, for which the maximum coverage was the same in the wild type and in Δsmc strain. This analysis showed that in such a case, there is still the difference in the spreading around *parS* dependent on the presence of SMC. We have now added this analysis as **Fig. S10**.

4. The authors do not take into account the possibility that the lower occupancy of *parS* sites in the Δsmc mutant could result from defects in chromosome organization that may affect the rate of ParB clamp loading or the postulated bridging activity of ParB.

Actually, we have suggested this mechanism as one of the explanations of SMC impact on ParB binding to *parS* sites (Discussion, lines 319- 321*).

5. The effect of SMC on the CTPase activity of ParB is relatively moderate. Moreover, the number of SMC molecules per cell is very low (~80 in *B. subtilis*) and these molecules are distributed across the entire chromosome. ParB, on the other hand, is typically significantly more abundant and condensed in segrosome complexes. Therefore, most ParB molecules would likely remain unaffected by SMC. It is therefore not clear how a direct interaction between SMC and ParB can determine the global dynamics of segrosomes, as proposed by the authors in their final model.

In our initial CTPase activity assays, we used a 1:10 molar ratio of ParB to SMC. However, in response to this comment, we tested lower SMC concentrations, using a ParB to SMC ratio of 100:1. This showed a similar impact of SMC on ParB - the results of these experiments are now presented in **Fig. 6**. Additionally, we are also showing that the ParB variant that is deficient in SMC binding exhibit only slightly reduced CTPase activity in the presence of SMC (**Fig. 6**). We believe that this result supports the role of direct ParB-SMC interactions in regulation of CTP hydrolysis by ParB and its CTP-hydrolysis dependent spreading.

Other issues:

6. The introduction is, in part, unstructured and does not provide the general reader with a clear explanation of the background knowledge required to understand the topic. The authors should introduce the formation of segrosomes (with the proper literature cited), the mode of action of ParABS systems (including a brief description of the segregation mechanism), the composition and role of SMC proteins, and the previous knowledge about the interplay between ParA, ParB and SMC before explaining the specific functions of these proteins in *Streptomyces* species.

We have reorganized and modified the Introduction. We have added some background information on the mechanism of chromosome segregation and the role of SMC.

7. It appears that the different strains shown in Figure S1A accumulate ParB-HT to quite different levels. Why is the level in the absence of aTet higher than in the presence of the inducer? The Western blot analysis performed with an anti-ParB antibody should include the wild-type strain without any deletions or tags. Otherwise, it is difficult to assess whether the levels of the tagged

proteins are indeed similar to that in the wild-type strain. Please indicate the antibodies used in the Figure.

In strains, KP008 (WT $p_{rtet}parB-HT$), KP009 ($\Delta parB p_{rtet}parB-HT$), KP010 ($\Delta parB::apra p_{rtet}parB-HT$) and KP011 ($\Delta parB p_{rtet}parB-HT, ftsZ-ypet$) $parB-HT$ gene was under the control of p_{rtet} promoter ($p_{tcp}830$ controlled by the reverse TetR repressor (TetR_{rv}) which binds to tetO sites in presence of tetracycline (Klotzsche et al., 2009) – thus ATET lowers the levels of ParB-HT. We have added this information to the strain-description in **Supplementary information and in Fig. S1 legend**. In Fig. S 1A the last lane represents ParB levels in wild type strain. We have added the information of antibodies used at the panel.

8. The Western blots in Fig. S6A show that there are not marked changes in the levels of ParB-HT over the course of sporogenic development in the wild-type and Δsmc backgrounds. However, since the two strains are analyzed on two different gels, it is not possible to conclude from these results that the lack of SMC does not change the ParB-HT levels.

The levels of ParB in the wild type strain and in $\Delta parB$ strain are compared in the Western blotting shown in **Fig. S6C**.

9. How do the authors explain the observation that, in vegetative Δsmc cells, the immobile ParB-HT molecules are not those forming the apical segrosome complex? Are the proteins no longer clamped onto the DNA in this complex?

The potential explanations for this indeed intriguing phenomenon are provided in the **Discussion** (lines 330-342*).

10. Faster segrosome disassembly in the absence of SMC should lead to fainter segrosome foci. The authors should measure the average intensity of segrosome complexes in the wild-type and Δsmc mutant to test this prediction.

Our other studies showed that lowered ParB spreading due to abolished CTP binding impairs the formation of ParB complexes or affects their size (Szymczak et al. under revision, <https://doi.org/10.1101/2024.12.04.625172>). The absence of SMC did not visibly affect the size and intensity of the foci (as shown in Fig S1C).

Comments/suggestions:

- Line 39: "Turnover of segrosomes" or "assembly and disassembly kinetics of ParB" - **corrected**
- Line 43: "several" instead of "numerous" - **ok**
- Line 52: "bacterium" –we refer to the bacterial genus here, so we use the plural
- Lines 72/73: Perhaps "in the model species *S. venezuelae*, which allows ..." - **modified**
- Line 96: "were followed" - **ok**

- Line 97ff: "fluorescence microscopy" - corrected
- Line 101: "diffused" is not the correct word here. Perhaps "gradually disassembled and completely disappeared" modified
- Line 134: Perhaps clearer: "the more rapid disassembly" - ok
- Figure 3A: The x-axes should be labeled in the graphs shown. - corrected
- Figure 3B: Did the authors use a global definition of the diffusion constants in the different strains or conditions? If so, this should be indicated. In this figure static and mobile population diffusion coefficient was calculated globally for all analyzed strains and timepoints in order to make comparisons between strain clearer.
- Figure 4: The proper term for Tau is "recovery half-time" - added
- Lines 137ff: The authors frequently refer to the diffusion constant, which averages the mobility of all molecules analyzed) or step sizes when reporting differences in protein mobility. The description of their results would be much clearer if they stated at the beginning that the diffusion behavior of ParB-HT suggests the existence of two distinct populations: an immobile population that is likely associated with DNA in segrosomes, and a mobile population that is likely freely diffusing. Then, the arguments should be made based on changes in the fraction sizes of these two populations.
We have added this explanation (Results, lines 159-162)
- Line 181: "MSD" is the abbreviation of "mean square displacement" – corrected
- Figure 3F: Please show the cell outlines. Figures in this plot were created only for the tip-proximal region of Streptomyces hyphae - 3µm from the hyphal tip. SMTracker does not allow the user to set which part of the analyzed cell is the hyphal tip, which means that it is only possible to check whether ParB localizes centrally within this 3 µm region. Since this is only a fragment of the entire hyphae it is not possible to add a cell outline. We have expanded the figure legend to clarify this.
- Line 224: This has also been shown for other ParB homologs. – corrected
- Line 307: "corroborates the results of earlier studies" - ok
- Line 312ff: The segrosomes are still equally spaced in the Δsmc mutant (and even earlier than in the wild type), suggesting that chromosome segregation is not affected.
Indeed, our analyses show that the chromosome segregation defect in \$\Delta smc\$ strain is modest (4% of anucleate spores) (Fig. S1D), however, it is significant in \$\Delta parB\Delta smc\$ double mutant (11% of anucleate spores).
- The references should be properly formatted and checked for errors and missing information.
The references have been corrected.

- Methods: How were the Western blots performed? Did the authors use equal volumes of culture or normalize to OD when preparing the samples? This makes a difference when assessing changes in the concentration of proteins over time.

Samples were prepared by mixing equal amounts of total protein (20 µg) from each lysate with 6× SB buffer. That explanation was now added to Supplementary Information.

***Line number in the file “Pawlikiewicz_04.2025 ms_changes tracked”**

We thank the Reviewers for approving the modifications introduced in the manuscript. We also appreciate Reviewer #4's discussion of a possible alternative explanation of SMC's impact on ParB. Inspired by the Reviewer's suggestions, we propose a slightly modified model of SMC positive feedback on the ParB complex. We address the suggestions and questions raised by the Reviewer below.

Reviewer #4 (Remarks to the Author):

In the revised version of the manuscript, the authors have performed several additional experiments to support their conclusions. Most importantly, they have generated a variant of ParB with amino acid exchanges in the putative SMC binding site to verify that the SMC-mediated decrease in the CTPase activity of ParB depends on a direct interaction between the two proteins. Although the additional data further support the hypothesis that SMC in some way affects segrosome assembly, there are still some issues that should be addressed:

1. The new data provided in Figure S10 do not address the previous question of whether the overall shapes of the ParB distributions obtained for WT and Δ smc cells at individual parS sites are similar. A comparison of distributions measured at different parS sites is not valid, because the sliding behavior of ParB is affected by different factors such as the presence roadblocks. If the WT (red) and Δ smc (blue) distributions in panels A are normalized so that their maxima at position 0 bp are identical, then the shapes and the width of the distributions are comparable. The same is true for the ChIP-seq peaks at other parS sites. This suggests that ParB is not less stably associated with the centromeric region after release from parS but rather that less ParB is loaded at parS in the first place. This would mean that SMC does not affect complex stability and spreading (as suggested in the abstract and Discussion) but rather the loading step.

We acknowledge the critical comment that the comparison of ParB binding to two parS sites has some limitations and does not fully support the conclusions on the differences in the ParB spreading. We therefore decided to remove Fig. S10.

We now revisited ChIP-seq results and reconsidered the diminished loading at parS in the absence of SMC. It should be noted that we showed earlier that ParB is a prerequisite for SMC loading. Thus we conclude that ParB is loaded on parS and then recruits SMC. However, we also take into account that the CTP hydrolysis assay performed in the absence and the presence of SMC suggest that the ParB molecules that spread along DNA in the presence of SMC have diminished CTPase activity. If SMC would only promote ParB loading at parS and closing the ParB clamp, the lack of SMC would considerably lower the number of ParB molecules spreading on DNA. In such a case, in the absence of SMC, the smaller number of

ParB molecules would exhibit CTPase activity resulting in lower detection of inorganic phosphate. That contradicts our observation suggesting that SMC affects ParB not only at the loading stage but also stabilises spreading protein.

Therefore, reevaluating all the data we propose the alternative model (Fig.7) in which SMC loop extrusion is critical for ParB complex stabilization. The SMC recruited by ParB, by extruding the DNA loops promotes the binding of ParB molecules to distant parS sites and by promoting distant ParB-ParB interactions (ParB-bridging) by NTD stabilizes the ParB clamps on DNA.

We believe that this model (presented in the new Fig.7) properly addresses our experimental data and acknowledges the Reviewer's suggestions. We include the model description in the Discussion.

2. The authors now show that SMC needs to directly interact with ParB to affect its CTPase activity in vitro. However, I still find it difficult to imagine how a small number of SMC complexes distributed throughout the chromosomal DNA (and thus spatially separated from ParB) are able continuously affect the stability of several hundred ParB clamps sliding in the centromeric regions by means of direct protein-protein interactions (as suggested as the most likely mechanism in the Discussion). It appears more likely that SMC affects the behavior of ParB specifically at the loading stage, thereby modulating the overall number of ParB clamps associated with the centromeric region. Consistent with this notion, the observation that SMC affects the ParB CTPase activity at substoichiometric concentrations suggests a catalytic effect rather than a stable interaction. Would it be possible that SMC is loaded by closed ParB clamps and SMC thus promotes the transition of ParB to the closed state at parS? It is difficult to tell whether the relatively small decrease in the CTPase activity observed in vitro is really physiologically relevant and not caused by the free diffusibility of ParB and SMC in solution, which enables frequent contacts throughout the CTPase cycle (in contrast to the in vivo situation). Generally, it is difficult to draw firm conclusions from this result without knowing whether/how the amino acid exchanges in the putative SMC binding site affect the overall CTPase cycle of ParB or the functionality of ParB in vivo.

We agree that the interaction between SMC and ParB may not be stable, which explains the small number of SMC molecules sufficient to stabilize ParB. However, we have confirmed that direct interaction is necessary to observe the impact of SMC on ParB CTPase activity. We believe that our verified model addresses Reviewers' questions - if upon recruitment of SMC by ParB, the SMC-dependent loop extrusion stabilizes the ParB complex by promoting the interactions by N-terminal domains the small number of SMC molecules will be sufficient to influence the ParB CTPase activity.

3. The authors relate the fast recovery of ParB complexes to the higher levels of ParB dimers that are non-specifically associated to chromosomal DNA. At the same time, they argue that this fast recovery indicates a shortened life-time of ParB complexes. Would it be possible that the lack of SMC leads to less ParB loading at *parS* and thus to a higher level of non-specifically associated ParB, which may be less stably bound and thus, in turn, lead to faster recovery?

*If in the absence of SMC just less ParB would be loaded on DNA, we should expect to see the smaller ParB foci in the absence of SMC, which is not the case. We rather suggest that in the absence of SMC the higher ParB turnover is due to the “loosened” complex. That is also consistent with the increased CTPase activity, which reflects a higher turnover of ParB loaded on *parS* and spreading. Rebinding of ParB to *parS* may be promoted by ParA (as suggested by our earlier data, Donczew, 2016).*

4. How do the authors explain the fact that the intensity of ParB foci does not differ between WT and Δ smc cells?

*We suppose that the model in which the absence of SMC destabilizes ParB molecules which are spreading along DNA, increasing turnover of the complexes is in agreement with the lack of the difference in the foci intensity. In contrast, if only ParB loading on *parS* would be inhibited in the absence of SMC, as suggested by the Reviewer, then we should observe less intensive foci.*

5. I still think that it is not justified to draw a direct connection between differences in the short-term kinetics of segrosome assembly and the long-term stability of these complexes over the course of the developmental cycle. Segrosomes turn over many times during this cycle, so that changes in the equilibrium level of ParB within segrosomes should be dependent on factors other than the absence or presence of the SMC-ParB interaction (even though lower equilibrium levels of ParB in the segrosomes may make such additional regulatory effects more easily detectable).

It should be considered that a higher turnover of ParB molecules involved in segrosome formation in a strain lacking SMC will promote a higher turnover of segrosomes. Higher turnover of ParB is consistent with increased detection of CTP hydrolysis in the absence of ParB than in its presence. If segrosomes undergo high turnover, they are expected to reassemble as long as the other developmental factors promote their association. As we suggested before we believe ParA may play the role at this stage. Segrosomes formed by ParB

stabilized by SMC are likely to last longer independently of diminishing levels of developmental factors (ParA).

- Line 304: "CTP binding" - *corrected*